# SAMG: Offline-to-Online Reinforcement Learning via State-Action-Conditional Offline Model Guidance

## Abstract

Offline-to-online (O2O) reinforcement learning (RL) pre-trains models on offline data and refines policies through online fine-tuning. However, existing O2O RL algorithms typically require maintaining the tedious offline datasets to mitigate the effects of out-of-distribution (OOD) data, which significantly limits their efficiency in exploiting online samples. To address this deficiency, we introduce a new paradigm for O2O RL called **S**tate-**A**ction-Conditional Offline **M**odel **G**uidance (SAMG). It freezes the pre-trained offline critic to provide compact offline understanding for each state-action sample, thus eliminating the need for retraining on offline data. The frozen offline critic is incorporated with the online target critic weighted by a state-action-conditional coefficient. This coefficient aims to capture the offline degree of samples at the state-action level, and is updated adaptively during training. In practice, SAMG could be easily integrated with Q-function-based algorithms. Theoretical analysis shows good optimality and lower estimation error. Empirically, SAMG outperforms state-of-the-art O2O RL algorithms on the D4RL benchmark.

## 1 Introduction

Offline reinforcement learning (Lowrey et al., 2019; Fujimoto et al., 2019; Mao et al., 2022; Rafailov et al., 2023) has gained significant popularity due to its isolation from online environments. It relies exclusively on offline datasets, which can be generated by one or several policies, constructed from historical data, or even generated randomly. This paradigm eliminates the risks and costs associated with online interactions and offers a safe and efficient pathway to pre-train well-behaved RL agents. However, offline RL algorithms exhibit an inherent limitation in that the offline dataset only covers a partial distribution of the state-action space (Prudencio et al., 2023). Therefore, standard online RL algorithms fail to resist the cumulative overestimation on samples out of the offline distribution (Nakamoto et al., 2023). To this end, most offline RL algorithms limit the decision-making scope of the estimated policy within the offline dataset distribution (Kumar et al., 2019; Yu et al., 2021). Accordingly, offline RL algorithms are conservative and are confined in performance by the limited distribution.

To overcome the performance limitation of offline RL algorithms and further improve their performance, it is inspiring to perform an online fine-tuning process with the offline pre-trained model. Similar to the successful paradigm of transfer learning in deep learning (Weiss et al., 2016; Iman et al., 2023), this paradigm, categorized as offline-to-online (O2O) RL algorithms, is anticipated to enable substantially faster convergence compared to pure online RL. However, the online fine-tuning process inevitably encounters out-of-distribution (OOD) samples laid aside in the offline pre-training process. This leads to another dilemma: the conservative pre-trained model may be misguided toward structural damage and performance deterioration when coming across OOD samples (Nair et al., 2020; Kostrikov et al., 2022). Therefore, O2O RL algorithms tend to remain unchanged or even sharply decline in the initial stage of the fine-tuning process. Existing algorithms conquer this by maintaining access to the offline dataset and retraining on the offline data during online iterations to restore offline information and restrict OOD deterioration.

Specifically, most fine-tuning algorithms directly inherit the offline dataset as online replay buffer and only get access to online data by incrementally replacing offline data with online ones through iterations (Lyu et al., 2022; Lee et al., 2022; Wen et al., 2024; Wang et al., 2024). This paradigm is tedious given that the sample size of the offline datasets tends to exceed the order of millions (Fu et al., 2020). Hence, these algorithms exhibit low inefficiency in leveraging online data. Other algorithms (Nakamoto et al., 2023; Zheng et al., 2023; Guo et al., 2023; Liu et al., 2024) maintain an online buffer and an offline one and sample from the two replay buffers with hybrid setting (Song et al., 2023) or priority sampling technique. Though these settings mitigate the inefficiency, they still visit a considerable amount of offline data and have not departed from the burden of offline data. In summary, existing algorithms severely compromise the efficiency of utilizing online data to mitigate the negative impact of OOD samples.

This compromise results in several undesirable outcomes. Training with offline data can potentially hindering algorithmic improvement given the sub-optimal nature of some offline data. Meanwhile, the inefficiency in accessing online samples limits the ability to explore and exploit novel information, making model improvement more challenging. In summary, this setting poses a challenge to the goal of the fine-tuning process to improve algorithm performance with limited training budget.

A recent work WSRL explores initializing the replay buffer in the online phase without retaining offline data (Zhou et al., 2024). However, WSRL takes a relatively straightforward approach of Q-ensemble techniques (Chen et al., 2021) to enhance algorithm generalization and resist distribution shift, a technique that inherently increases model complexity and computational overhead.

To tackle the challenge of low online sample utilization while not introducing excessive computational burdens, it is inspiring to directly leverage the offline critic, which is learned from the offline dataset, forming a compact abstraction of the offline information. To this end, this paper introduces a novel online fine-tuning paradigm named **S**tate-**A**ction-Conditional Offline **M**odel **G**uidance (SAMG), which eliminates the need for retaining offline data and achieves 100% online sample utilization. SAMG freezes the offline pre-trained critic, which contains the offline cognition of the values given a state-action pair and offers offline guidance for online fine-tuning process. SAMG combines the offline critic with online target critic weighted by a state-action-conditional coefficient to provide a compound comprehension perspective. The state-action-conditional coefficient represents a class of functions that quantify the offline confidence of a given state-action pair and is instantiated as a Conditional Variational Autoencoder (C-VAE) model. It is adaptively updated during training to provide accurate probability estimation. SAMG only introduces minimal computational overhead while achieving excellent performance. It avoids introducing inappropriate intrinsic rewards by leveraging this probability-based mechanism. It does not affect offline algorithms and can be easily deployed on Q-function-based RL algorithms, demonstrating strong applicability.

The main contributions of this paper are summarized as follows: (1) The tedious offline data is eliminated to facilitate more effective online sample utilization. (2) The compact offline information generated by offline model is integrated to provide offline guidance. A novel class of state-action-conditional function is designed and updated to estimate the offline confidence. (3) Rigorous theoretical analysis demonstrates good convergence and lower estimation error. SAMG is integrated into four Q-learning-based algorithms, showcasing remarkable advantages.

## 2 PRELIMINARIES

**Reinforcement learning** task is defined as a sequential decision-making process, where an RL agent interacts with an environment modeled as a Markov Decision Process (MDP): $\mathcal{M} = (\mathcal{S}, \mathcal{A}, P, r, \gamma, \tau)$. $\mathcal{S}$ represents the state space and $\mathcal{A}$ represents the action space. $P(s'|s, a)$ denotes the unknown function of transition model and $r(s, a)$ denotes the reward model bounded by $|r(s, a)| \leq R_{max}$. $\gamma \in (0, 1)$ denotes the discount factor for future reward and $\tau$ denotes the initial state distribution. The goal of the RL agent is to acquire a policy $\pi(a|s)$ to maximize the cumulative discounted reward, defined as state-action value function $Q^\pi(s, a) = \mathbb{E}_\pi \left[ \sum_{k=0}^\infty \gamma^k r(s_k, a_k) | s_0 = s, a_0 = a \right]$. The training process for actor-critic algorithms alternates between policy evaluation and policy improvement phases. Policy evaluation phase maintains an estimated Q-function $Q_\theta(s, a)$ parameterized by $\theta$ and updates it by applying the Bellman operator: $\mathcal{B}^\pi Q \doteq r + \gamma P^\pi Q$, where $P^{\pi_\phi} Q(s, a) = \mathbb{E}_{s' \sim P(s'|s,a), a' \sim \pi_\phi(a'|s')} \left[ Q(s', a') \right]$. In the policy improvement phase, the policy $\pi_\phi(a|s)$ is parameterized by $\phi$ and updated to achieve higher expected returns.

## 3 SAMG: METHODOLOGY

In this section, we introduce the SAMG paradigm, which leverages the pre-trained offline model to guide the online fine-tuning process without relying on tedious offline data. This approach raises three key questions: (1) How can we accurately extract the information contained within the offline model? (2) How can we assess the reliability of this information? (3) How can we adaptively adjust the level of reliability throughout the training process? To resolve these challenges, We propose a novel model-guidance technique and introduce an adaptive state-action-conditional coefficient.

### 3.1 OFFLINE-MODEL-GUIDANCE PARADIGM

Offline-model-guidance paradigm is designed to address Problem 1. Intuitively, the offline pre-trained value function $Q_\theta^{off}(s, a)$ of an algorithm estimates the quality of a specific state-action pair in the perspective of the offline dataset. This well-trained offline Q-network can be frozen and preserved to provide offline opinion when encountering online state-action pairs. To leverage both offline and online sights, the frozen offline Q-values are integrated with online Q-values weighted by a state-action-conditional coefficient. This approach brings several advantages: it can adaptively utilize the offline information based on its reliability and mitigate the introduction of undesirable intrinsic rewards, which will be discussed later. Formally, the policy evaluation equation is as follows:

$$Q(s,a) = r(s,a) + \gamma \left[ (1 - p(s,a))Q(s',a') + p(s,a)Q^{off}(s',a') \right]. \tag{1}$$

where $Q(s, a)$ represents the estimated Q-function and $p \in (0, 1)$ denotes a function class that gives a state-action-conditional coefficient and could be implemented with any reasonable form. The novel parts of the equation compared to the standard Bellman equation are marked in blue.

### 3.2 STATE-ACTION-CONDITIONAL COEFFICIENT

State-action-conditional coefficient is proposed to address Problem 2. Intuitively, we tend to allocate higher values to samples within the offline distribution, as these samples are well-represented in the offline data and the model is thoroughly pre-trained on them. Conversely, we have limited knowledge about samples distant from the offline distribution (treated as OOD samples), so lower values are appropriate. In summary, the state-action-conditional coefficient should capture the offline confidence of given samples, which resembles the role of behavior policy in offline RL (Prudencio et al., 2023). This coefficient attempts to depict the characteristics of the complex distribution represented by the offline dataset. Any structure that satisfies the criteria can serve as an instantiation of $p(s, a)$. However, considering the high-dimensional and continuous property of the state-action data, it is challenging to directly extract the probability characteristics from the state-action pair.

In this work, we adopt the C-VAE model to instantiate $p(s, a)$. C-VAE is a generative model designed to capture complex conditional data distributions by incorporating additional information. It can properly approximate the behavior policy and capture the underlying structure by introducing conditional variables such as actions or states. Therefore, it is widely used to estimate the behavior policies in offline RL (Fujimoto et al., 2019; Kumar et al., 2019; Xu et al., 2022; Guo et al., 2023). Its encoder $\text{Enc}_{\psi_1}$ maps the input data to the mean $z_m$ and variance $z_v$ parameters of a Gaussian distribution $\mathcal{N}(z_m, z_v)$. Latent vector $z$ is then sampled from this estimated distribution and then fed to the decoder $\text{Dec}_{\psi_2}$ to reconstruct the data (Kingma et al., 2014). The $\mathcal{N}(z_m, z_v)$ extracted from the encoder represents a lower-dimensional representation of the offline data distribution, which not only facilitates coefficient approximation but also enables OOD detection.

Nevertheless, previous work has mainly focused on the quality of the generated data, with limited attention to whether the distribution $\mathcal{N}(z_m, z_v)$ carries meaningful information. Consequently, the latent distribution tends to collapse towards the standard normal distribution due to the KL-divergence regularization, and $z$ is meaningless—a phenomenon known as **posterior collapse** (Lucas et al., 2019; Wang et al., 2021), as evidenced in Appendix C.1. This phenomenon is detrimental in our setting because we need the latent output to calculate the state-action-conditional coefficient. However, under posterior collapse, the model fails to function and the sampled latent variable $z$ is only normal noise.

To mitigate the adverse impacts of posterior collapse, we extend the variational conditional information to include state-action pairs and reconstruct the next state from the decoder. This approach complicates the modeling process and develops a state-action-conditional structure. Additionally,

we employ the KL-annealing technique (Bowman et al., 2015) to further alleviate posterior collapse, with a detailed explanation in Appendix C.2. Formally, the C-VAE component of SAMG is trained by optimizing the evidence lower bound (ELBO) objective function as commonly used in the C-VAE frameworks:

$$\max_{\psi_1, \psi_2} \mathbb{E}_{z \sim \text{Enc}_{\psi_1}} \left[ \log \text{Dec}_{\psi_2}(s'|z, s, a) \right] - \beta D_{KL} \left[ q_{\text{Enc}}(z|s, a) || p_{prior}(z) \right] \tag{2}$$

where $\text{Enc}_{\psi_1}(z|s, a)$ and $\text{Dec}_{\psi_2}(s'|z, s, a)$ represent the encoder and decoder structure respectively; $\text{Dec}_{\psi_2}(z)$ denotes the prior distribution of the encoder; and $D_{KL}[p||q]$ denotes the KL divergence. The former error term denotes the reconstruction loss while the latter denotes the KL divergence between the encoder distribution and the prior distribution.

### 3.3 COEFFICIENT GENERATION AND ADAPTIVE UPDATES

#### STATIC COEFFICIENT GENERATION

To validate the effectiveness of improved C-VAE structure, we evaluate the offline dataset by inputting each sample to the trained C-VAE model and recording the mean and variance values of encoder output. The result is illustrated in the Appendix C.3. The results indicate that posterior collapse is significantly alleviated. However, the normal distribution of encoder output $\mathcal{N}(z_m, z_v)$ is still relatively narrow. It is unreliable to directly utilize the latent information $z$ which is sampled from this narrow distribution because the sampling randomness may overshadow the distribution information.

To address this issue, we resort to utilize the deterministic information of $(z_m, z_v)$ in place of less reliable $z$. Because we have collected a sufficient number of $(z_m, z_v)$ from the offline dataset, we can fit the distribution of these samples. This fitted distribution can then serve as a representation of the offline dataset distribution. Since the statistical distributions of $(z_m, z_v)$ on offline dataset closely approximates a normal distributions, as evidenced by the minimal fitting error in Appendix C.3, we fit these samples to the corresponding normal distributions. Specifically, $z_m$ is modeled as $\mathcal{N}(\mu_m, \sigma_m)$, denoted as $Z_m$, while $z_v$ is modeled as $\mathcal{N}(\mu_v, \sigma_v)$, denoted as $Z_v$. Accordingly, for some observed sample $(z_m, z_v)$, we compute the probability that $Z_m$ ($Z_v$) falls within the same distance from the mean as $z_m$ ($z_v$) as shown below. Refer to Appendix C.4 for the complete derivation.

$$\begin{aligned} P(|Z_m - \mu_m| > |z_m - \mu_m|) &= 2F_{Z_m}(\mu_m - |z_m - \mu_m|) \\ P(|Z_v - \mu_v| > |z_v - \mu_v|) &= 2F_{Z_v}(\mu_v - |z_v - \mu_v|) \end{aligned} \tag{3}$$

where $F_X(x)$ is the cumulative distribution function. The intermediate probability can be obtained:

$$p^{int}(s, a) = \omega P(|Z_m - \mu_m| > |z_m - \mu_m|) + (1 - \omega)P(|Z_v - \mu_v| > |z_v - \mu_v|) \tag{4}$$

where $\omega$ is the weight of mean and standard and is set to 1 because the estimation error of the mean is significantly smaller than that of the standard in practice.

Moreover, in cases where the sample diverges notably from the offline distribution, the information about the sample is unknown and the offline guidance may be biased. Such samples are considered as OOD samples. To identify these samples, we use the intermediate probability $p^{int}$, which quantifies the probability of a sample belonging to the offline distribution. Specifically, samples with $p^{int}$ below $p_m^{off}$ are regarded as OOD samples, with a threshold probability $p_m^{off}$ introduced.

The eventual equation to calculate the probability $p^{off}$ given a sample $(z_m, z_v)$ is illustrated below:

$$p^{off}(s, a) = \begin{cases} p^{int}(s, a), & p^{int}(s, a) \geq p_m^{off} \\ 0, & p^{int}(s, a) < p_m^{off} \end{cases} \tag{5}$$

By integrating the C-VAE form state-action-conditional coefficients into Eq. (1), the following practical updating equation can be obtained and the structure of SAMG is illustrated in Fig. 1.

$$Q(s, a) = r(s, a) + \gamma \left[ (1 - p^{off}(s, a))Q(s', a') + p^{off}(s, a)Q^{off}(s', a') \right]. \tag{6}$$

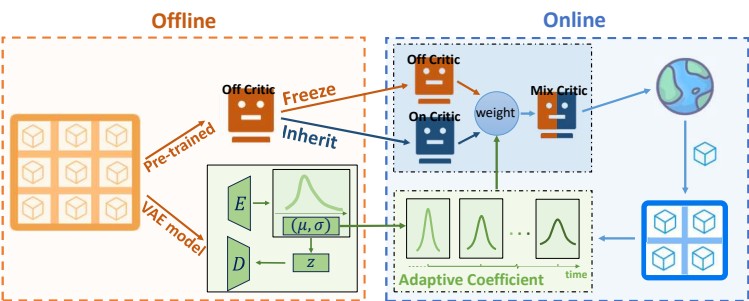

Figure 1: **Architecture of SAMG**. This figure illustrates the structure of SAMG, highlighting the transition from offline pre-training to online fine-tuning. It outlines key components, including offline critic, VAE model, offline-guidance technique, and adaptive coefficient.

### ADAPTIVE COEFFICIENT UPDATES

As the online fine-tuning processes, the agent's understanding of OOD samples evolves. Samples initially considered OOD by the static VAE might become well-understood by the online agent. To maintain the relevance of $p^{off}$, we propose an adaptive update mechanism for the VAE (Problem 3). Periodically, online samples initially deemed OOD samples ($p^{off} < p_m^{off}$) are re-evaluated. Those samples now mastered by the online agent (exhibiting low online Bellman error) are used to fine-tune the VAE model. This refinement allows $p^{off}$ to better reflect the agent's current capabilities in modulating the guidance. Refer to Appendix C.5 for complete implementation. The process of Section 3.3 is depicted in the green region of Fig. 1.

## 4 ANALYSIS OF SAMG

### 4.1 INTRINSIC REWARD ANALYSIS OF SAMG

Intrinsic Reward Analysis highlights the importance of the probability-based coefficient paradigm. Specifically, Eq. (6) can be derived as below:

$$Q(s, a) = [r(s, a) + r^{in}(s, a)] + \gamma Q(s', a'). \tag{7}$$

where $r^{in}(s, a) = \gamma p(s, a)(Q^{off}(s', a') - Q(s', a'))$. Eq. (7) indicates that the introduced offline information could be treated as the intrinsic reward.

Previous work has revealed that intrinsic reward may cause training instability or even algorithm degradation (Chen et al., 2022; McInroe et al., 2024). However, the intrinsic reward form of SAMG is reasonable and stable thanks to the probability-shape coefficient. Specifically, the intrinsic reward term describes the difference between offline and online Q-values, weighted by the state-action-conditional coefficient. It can be analyzed in two scenarios. Firstly, if the state-action pair lies within the offline distribution (ID), where the offline Q-value is well trained and the state-action-conditional coefficient $\alpha$ is significant. For the ID condition, although $Q$ is initialized by $Q^{off}$, due to the challenges of O2O training, $Q$ may be significantly affected and thus deviate from the correct value for ID samples. In this case, this term suggests that higher offline Q-values correspond to higher potential returns. Hence, it encourages exploring state-action pairs with higher performance. Conversely, if the state-action pair falls outside the offline distribution, where the offline Q-value may be erroneously estimated. This term becomes negligible or is even set to zero, as specified in Eq. (5). Therefore, it can filter out inaccurate and unreliable information. In summary, SAMG is able to properly retain the offline knowledge without introducing inappropriate intrinsic rewards.

Moreover, the intrinsic reward term is directly based on the Q-function, offering long-horizon guidance that is directly grounded in the function itself, offering a more temporally coherent learning signal.

## 4.2 THEORETICAL ANALYSIS OF SAMG

In this section, we adopt the temporal difference paradigm (Sutton, 1988; Haarnoja et al., 2018) in the tabular setting and prove that Eq. (6) still converges to the same optimality, even with an extra term induced. For the theoretical tools, SAMG gets rid of the offline dataset and therefore diverges from the hybrid realm of Song et al. (Song et al., 2023) and offline RL scope limited by the dataset, but aligns with online RL algorithms (T. Jaakkola & Singh, 1994; Thomas, 2014; Haarnoja et al., 2018).

**Contraction Property** is considered and proven to still hold since the Bellman operator is modified (Keeler & Meir, 1969). The related theorem and detailed proof can be found in Appendix B.1.

**Convergence Optimality.** Formally, iterative TD updating form of Eq. (6) is demonstrated below:

$$Q_{k+1}(s,a) - Q_k(s,a) = \alpha_k(s,a) \left[ Q_k(s,a) - (r_{k+1} + \gamma Q_k(s_{k+1}, a_{k+1})) \right] - \\ \alpha_k(s,a)\gamma p(s,a) \left( Q^{off}(s_{k+1}, a_{k+1}) - Q_k(s_{k+1}, a_{k+1}) \right) \tag{8}$$

where the estimated state-value function at time-step $k$ for given $(s,a)$ pair is denoted as $Q_k(s,a)$. The learning rate at time-step $k$ is represented as $\alpha_k$. For simplicity, the state-action conditional coefficient $p^{off}(s,a)$ and threshold $p_m^{off}(s,a)$ are denoted as $p(s,a)$ and $p_m(s,a)$, respectively.

Our theoretical analysis first focuses on the convergence properties of the policy evaluation step under the modified Bellman operator. We show that for any fixed policy $\pi$, the Q-values estimated using SAMG's update rule converge to the true $Q^\pi(s,a)$. For the complete proof, refer to Appendix B.2.

**Theorem 4.1** (Convergence property of SAMG). *For a given policy $\pi$, by the TD updating paradigm, $Q_k(s,a)$ of SAMG converges almost surely to $Q^\pi(s,a)$ as $k \to \infty$ for all $s \in \mathcal{S}$ and $a \in \mathcal{A}$ if $\sum_k \alpha_k(s,a) = \infty$ and $\sum_k \alpha_k^2(s,a) < \infty$ for all $s \in \mathcal{S}$ and $a \in \mathcal{A}$.*

**Convergence Speed.** Moreover, the specific expression for the contraction coefficient is proven as follows, illustrating the faster convergence speed of SAMG. See Appendix B.3 for further details.

**Theorem 4.2** (Convergence speed of SAMG). *The Bellman operator of SAMG satisfies the contraction property $\left\| \mathcal{B}(x) - \mathcal{B}(y) \right\| \leq \gamma_o \left\| x, y \right\|$ for all $x, y \in \mathcal{Q}$. $\mathcal{Q}$ represents the Q function space. The contraction coefficient of SAMG $\gamma_o(s,a)$ is bounded above by the following expression:*

$$\begin{cases} (1 - p(s,a))\gamma + \gamma \gamma_\mathcal{F} p(s,a)C, & p(s,a) \geq p_m \\ \gamma, & p(s,a) < p_m \end{cases} \tag{9}$$

*where $C = \left\| \Delta_{off}(s,a) \right\|_\infty / \left\| \Delta_k(s,a) \right\|_\infty$ denotes the ratio of the offline and online suboptimality bounds, $\left\| \Delta_{off}(s,a) \right\|_\infty$ denotes the offline suboptimality bound $[V^*(s) - V^{\pi_{off}}(s)]$, $\left\| \Delta_k(s,a) \right\|_\infty$ denotes the suboptimality bound of the k-th iteration of online fine-tuning and $0 < \gamma_\mathcal{F} < 1$ denotes the convergence coefficient of offline algorithm class $\mathcal{F}$.*

The upper equation of Theorem 4.2 holds for in-distribution samples, which are well mastered by the offline model. Therefore, the offline suboptimality bound is substantially tighter compared to the online bound. This illustrates that the offline model guidance significantly accelerates the online fine-tuning process by providing more accurate estimations for in-distribution samples. Specifically, for these samples, the convergence speed depends on the offline confidence implied by $p(s,a)$, i.e., a higher $p(s,a)$ indicates a higher degree of offline-ness, corresponding to a smaller error term constrained by the term $(1 - p(s,a))$ and ensuring faster convergence. For the OOD samples, the algorithm degenerates into the traditional algorithm because $p(s,a)$ is set to zero as defined in Eq. (5). This theoretical result is highly consistent with the analysis of the expected performance as stated in Section 4.1. Furthermore, it indicates that the extent of algorithm improvement is influenced by the sample coverage rate of the offline dataset. Specifically, the offline guidance is more reliable with more complex sample coverage, whereas the guidance is constrained with limited sample diversity.

## 5 EXPERIMENTAL RESULTS

Our experimental evaluations focus on the performance of SAMG during the online fine-tuning process based on three state-of-the-art algorithms within D4RL (Fu et al., 2020), covering diverse environments and task complexities, as detailed below.

Table 1: **Performance comparison.** The D4RL normalized score (Fu et al., 2020) is evaluated for standard base algorithms (including CQL (Kumar et al., 2020), IQL (Kostrikov et al., 2022) and AWAC (Nair et al., 2020), denoted as "Vanilla") in comparison to the base algorithms augmented with SAMG (referred to as "Ours"), as well as three baselines (TD3BC (Chen et al., 2020), SPOT (Wu et al., 2022), Cal_QL (Nakamoto et al., 2023) and EDIS (Liu et al., 2024)). The superior scores are highlighted in  blue . The result is the average normalized score of 5 random seeds $\pm$ (standard deviation).

| Dataset[1] | CQL | | AWAC | | IQL | | TD3BC | SPOT | Cal_QL | EDIS |
|---|---|---|---|---|---|---|---|---|---|---|
| | Vanilla | Ours | Vanilla | Ours | Vanilla | Ours | | | | |
| Hopp-mr | 100.6(1.8) | 103.7 (1.3) | 99.4(1.3) | **108.3** (0.2) | 86.2(16.1) | 100.4 (0.9) | 64.4(21.5) | 68.0(11.2) | 80.9(38.2) | 83.0(26.8) |
| Hopp-m | 60.2(2.7) | 88.3 (6.0) | 88.2(14.6) | **102.5** (1.8) | 62.1(7.4) | 68.4 (2.9) | 66.4(3.5) | 54.6(7.1) | 78.1(8.7) | 30.1(8.9) |
| Hopp-me | 110.8(1.0) | **113.0** (0.3) | 101.9(20.5) | 112.8 (7.2) | 103.5(8.7) | 108.1 (3.1) | 101.2(9.1) | 82.6(11.5) | 109.1(0.2) | 78.4(3.5) |
| Half-mr | 48.0(0.5) | 57.8 (1.7) | 48.9(1.1) | **62.8** (3.3) | 45.1(0.6) | 49.6 (1.0) | 44.8(0.6) | 42.4 (3.7) | 51.6(0.8) | 82.9(1.2) |
| Half-m | 47.6(0.2) | 59.0 (0.7) | 54.2(1.1) | **69.5** (0.9) | 49.3(0.1) | 62.5 (1.5) | 48.1(0.2) | 45.9(2.4) | 63.2(2.5) | 66.4(11.7) |
| Half-me | 95.2(1.0) | **97.2** (0.8) | 94.8(1.3) | 96.7 (1.0) | 91.6(0.9) | 82.3(11.7) | 90.8(6.0) | 87.4(7.4) | 95.6(4.3) | 90.2(1.4) |
| Walk-mr | 82.7(0.7) | 88.4 (5.0) | 93.8(3.4) | **120.1** (3.1) | 87.1(3.3) | 99.5 (2.4) | 85.6(4.0) | 69.2 (6.2) | 97.1(2.5) | 46.9(23.6) |
| Walk-m | 60.2(2.7) | 82.9 (1.8) | 87.8(0.8) | **103.6** (4.5) | 83.4(1.6) | 88.6 (4.7) | 82.7(4.8) | 79.5(2.4) | 83.6(0.8) | 76.2(16.7) |
| Walk-me | 109.5(0.5) | 112.5 (0.7) | 112.7(0.9) | **129.2** (3.6) | 113.6(1.1) | 116.3 (3.7) | 110.0(0.4) | 87.8(3.9) | 110.7(0.4) | 107.9(10.3) |
| Ant-u | 92.0(1.7) | **97.0** (1.4) | 70.0(40.4) | 87.0 (13.2) | 83.3(6.1) | 94.0 (1.2) | 70.8(39.2) | 30.8(12.9) | 96.8(0.4) | 95.0(7.0) |
| Ant-ud | 58.0(32.0) | 62.4 (12.4) | 15.0(35.3) | 75.0 (7.0) | 33.2(4.4) | **77.8** (0.8) | 44.8(11.6) | 44.8(6.5) | 63.8(43.4) | 72.4(32.5) |
| Ant-md | 82.4(2.2) | 89.2 (3.5) | 0.0(0.0) | 0.0(0.0) | 76.4(5.4) | **96.6** (1.9) | 0.4(0.4) | 36.2(11.0) | 93.4(3.6) | 82.4(4.8) |
| Ant-mp | 85.6(6.6) | 86.4 (1.1) | 0.0(0.0) | 0.0(0.0) | 76.2(4.6) | **95.2** (1.6) | 0.4(0.4) | 38.4(8.7) | 94.0(2.2) | 60.0(51.9) |
| Ant-ld | 62.8(7.4) | 63.8 (5.9) | 0.0(0.0) | 0.0(0.0) | 45.4(7.7) | **81.4** (7.9) | 0.0(0.0) | 0.0(0.0) | 78.8(5.8) | 32.6(15.0) |
| Ant-lp | 55.0(8.4) | 60.8 (1.5) | 0.0(0.0) | 0.0(0.0) | 48.8(7.7) | **74.8** (8.4) | 0.0(0.0) | 0.0(0.0) | 73.0(19.4) | 35.0(17.3) |
| Pen-c | 90.0(4.6) | 96.2 (4.0) | 63.3(39.7) | 70.1 (26.6) | 86.7(24.6) | **106.0** (27.8) | 6.4(4.37) | 2.80(11.82) | -0.03(4.10) | 8.99(17.18) |
| Door-c | -0.34(0.01) | **70.8** (2.8) | 0.00(0.01) | 7.29 (3.14) | -0.06(0.03) | 13.08 (3.20) | -0.32(0.01) | -0.16(0.05) | -0.33(0.01) | 0.09(0.03) |
| Relo-c | -0.28(0.12) | **75.0** (2.1) | -8.84(1.22) | -7.82 (0.93) | -0.01(0.04) | 0.24 (0.03) | -0.21(0.01) | -0.14(0.10) | -0.31(0.03) | -0.34(0.02) |

[1] Hopp: Hopper, Half: HalfCheetah, Walk: Walker2d, Ant: Antmaze, Relo: Relocate, mr: medium-replay, me: medium-expert, d: diverse, p: play, u: umaze, ud: umaze-diverse, md: medium-diverse, mp: medium-play, ld: large-diverse, lp: large-play, c: cloned.

**Baselines.** (i) **SAMG algorithms**, the SAMG paradigm is constructed on a variety of state-of-the-art O2O RL algorithms, including CQL (Kumar et al., 2020) and AWAC (Nair et al., 2020), IQL (Kostrikov et al., 2022). The pseudo-code of SAMG is provided in Appendix D.1. (ii) **O2O RL algorithms**, we implement the aforementioned O2O RL algorithms (CQL, AWAC and IQL). We also implement SPOT (Wu et al., 2022) (iii) **Hybrid RL**, we implement SOTA hybrid-RL-based algorithms, including Cal_QL (Nakamoto et al., 2023) and EDIS (Liu et al., 2024). (iv) **Behavior Cloning (BC)**, we implement Behavior cloning based algorithm TD3+BC (Fujimoto & Gu, 2021). All algorithms are implemented based on CORL library (Tarasov et al., 2024) with implementation details in Appendix D.2. To ensure a fair comparison, all algorithms are pre-trained offline for **1M iterations** followed by **200k iterations** of online fine-tuning, which is significantly shorter than previous work.

**Benchmark tasks.** We evaluate SAMG and the baselines across multiple benchmark tasks: **(1)** The Mujuco locomotion tasks (Fu et al., 2020), including three different kinds of environments (HalfCheetah, Hopper, Walker2d) where robots are manipulated to complete various tasks on three different levels of datasets. **(2)** The AntMaze tasks that an "Ant" robot is controlled to explore and navigate to random goal locations in six levels of environments. **(3)** The Adroit tasks include Pen, Door, and Relocate environments. Details are stated in Appendix D.3.

## 5.1 EMPIRICAL RESULTS

The normalized scores of the vanilla algorithms with and without SAMG integrated are shown in Table 1. SAMG consistently outperforms the vanilla algorithms in the majority of environments, illustrating the superiority of SAMG. SAMG converges significantly faster than vanilla algorithms and can achieve higher performance. Notably, SAMG achieves the best performance with the simpler algorithm AWAC, while delivering substantial improvements with other algorithms. The reason for this counter-intuitive phenomenon is discussed in Appendix D.4, which just illustrates the

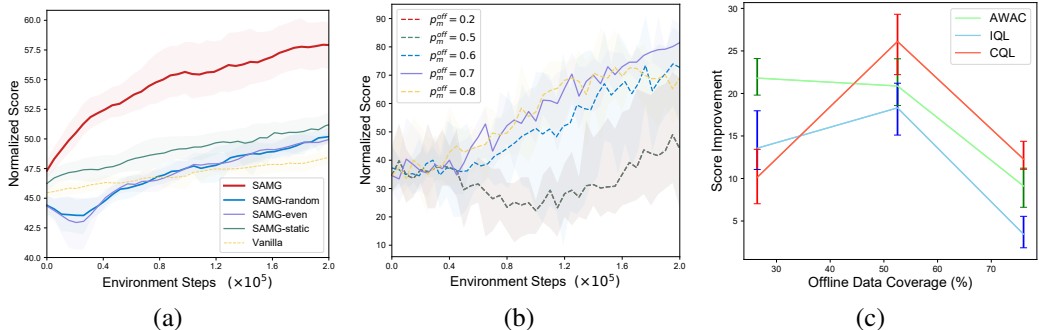

Figure 2: The left figure (a) illustrates ablation analysis of the state-action-conditional coefficient. The middle figure (b) demonstrates the sensitivity test for the coefficient threshold $p_m^{off}$ (Please note that the curve $p_m^{off} = 0.2$ and $p_m^{off} = 0.5$ are overlapped). The right figure (c) plots the average improvement of the normalized score over the offline data coverage.

effectiveness of SAMG. We present the cumulative regrets on Antmaze in Appendix D.5, further demonstrating the outstanding online sample efficiency of SAMG.

Although SAMG performs well in most environments, it is still worthwhile to notice SAMG may occasionally behave unsatisfactory (e.g., IQL-SAMG on HalfCheetah-medium-expert task). We discuss in Appendix D.6 that this exception is caused by the environment rather than the defect of SAMG. We notice that the AWAC algorithm performs poorly in the Antmaze environment, resulting in SAMG struggling to initiate. This is because AWAC is relatively simple and not competent for the complex task of Antmaze; it is an inherent limitation of AWAC, rather than an issue with SAMG.

We further **compare the offline-data-free algorithms** SAMG and WSRL in Appendix D.7, highlighting the superiority and contribution of SAMG.

## 5.2 ABLATION ANALYSIS OF THE COEFFICIENT

The state-action-conditional coefficient $p(s, a)$, instantiated as $p^{off}(s, a)$, estimates the offline degree for a given $(s, a)$ pair and is adaptively updated during training. To demonstrate the impact of this adaptive state-action-conditional coefficient, we compare several different architectures on the environment HalfCheetah with CQL-SAMG, including: (i) adopted SAMG setting (denoted as SAMG), (ii) static state-action-conditional coefficient (which means the VAE model is fixed once pre-trained offline, denoted as SAMG-static) (iii) the offline and online critics are combined with equal weights (0.5 each) for each state-action pair. (denoted as SAMG-even), (iv) the mixing coefficient for each state-action pair is randomly sampled from a uniform distribution (denoted as SAMG-random), (v) the vanilla RL algorithms (denoted as Vanilla). The results are illustrated in Fig. 2 (a).

SAMG shows consistent and significant improvement compared to other settings. Casual selections of C-VAE (SAMG-even and SAMG-random) exhibit notably inferior algorithm performance during the initial training phase, demonstrating the effectiveness of state-action-conditional coefficient structure. However, they catch up with and surpass the performance of the vanilla algorithms, highlighting the advantage of SAMG paradigm and offline information. SAMG improves over the SAMG-static and SAMG-even algorithms by **15.3% and 21.8%** respectively.

## 5.3 SENSITIVITY ANALYSIS OF COEFFICIENT THRESHOLD

This crucial hyperparameter, $p_m^{off}$, holds the lower threshold of OOD samples. We evaluate the sensitivity on Antmaze with IQL-SAMG across a range of numbers including 0.2, 0.5, 0.6, 0.7 (chosen value), and 0.8. The results shown in Fig. 2 (b) illustrates that the influence of $p_m^{off}$ is **systematic and interpretable**. A slight change in the value of $p_m^{off}$ (corresponding to 0.6 and 0.8) leads to an insignificant influence to the algorithm performance. If the value is too small, all samples are regarded as in-distribution samples and the $p^{off}(s, a)$ is identical to $p^{int}(s, a)$. Therefore the performance remains identical across different $p_m^{off}$, as evidenced by the overlapping curves for 0.2

and 0.5 in Fig. 2. In summary, tuning $p_m^{off}$ is controllable and the range of 0.6 to 0.8 is sufficient, and the performance remains relatively stable within this interval.

## 5.4 FURTHER COMPARISONS WITH HYBRID RL ALGORITHMS

We compare SAMG with Hybrid RL based algorithms in Table 1. A natural problem arises: would the algorithm performance improve if the hybrid RL setting were replaced with offline-guidance setting? To illustrate this question, we modify the Cal_QL algorithm with SAMG setting and eliminate the offline data. The results, presented in Table 2, indicate that SAMG still outperforms Cal_QL, demonstrating the superiority of SAMG paradigm. As for EDIS algorithm, it relies heavily on the offline dataset, making it impractical to adapt to offline-guidance setting.

## 5.5 DOES SAMG RELY ON OFFLINE DATA COVERAGE?

This part aims to showcase the relationship between algorithm performance improvement and the coverage rate of the offline dataset. To quantify the data coverage of a specific offline dataset, we apply t-SNE (Van der Maaten & Hinton, 2008) to perform dimensionality reduction and then cluster data points across all levels of datasets within a given environment, as detailed in Appendix D.8.

As shown Fig. 2 (c) (left: medium-replay, middle: medium and right: medium-expert), we observe consistent performance improvement of SAMG across all dataset levels. Notably, middle sample coverage rate yields more significant performance improvement. This is because extremely low coverage induces a narrow distribution of the offline dataset, resulting in limited information of the offline model. Conversely, high coverage contributes to satisfaction with the offline model, thus leaving limited room for further enhancement. Moreover, moderate coverage scenarios are common in practical offline datasets, making the observed behavior particularly relevant in real-world settings.

Table 2: **Algorithm performance of Cal_QL and SAMG.** The algorithms performance of Cal_QL compared to SAMG integrated Cal_QL algorithms. The setting and notions are the same as Table 1.

| | Hopp-mr | Hopp-m | Hopp-me | Half-mr | Half-m | Half-me | Walk-mr | Walk-m | Walk-me |
|---|---|---|---|---|---|---|---|---|---|
| Cal_QL | 80.9(38.2) | 78.1(8.7) | 109.1(0.2) | 51.6(0.8) | 63.2(2.5) | 95.6(4.3) | 97.1(2.5) | 83.6(0.8) | 110.7(0.4) |
| SAMG | 101.6(0.6) | 99.8(2.1) | 111.7(0.6) | 56.4(1.5) | 65.1(1.0) | 96.3(0.5) | 101.2(0.1) | 97.8(1.7) | 112.3(0.8) |
| | Ant-u | Ant-ud | Ant-mp | Ant-md | Ant-lp | Ant-ld | Pen-c | Door-c | Relo-c |
| Cal_QL | 96.8(0.4) | 63.8(43.4) | 93.4(3.6) | 94.0(2.2) | 78.8(5.8) | 73.0(19.4) | -0.03(4.10) | -0.33(0.01) | -0.31(0.03) |
| SAMG | 99.0(1.0) | 66.4(23.0) | 91.6(2.4) | 96.0(1.6) | 79.8(2.4) | 72.4(11.2) | 5.01(12.01) | 1.35(0.57) | 3.89(0.34) |

## 6 RELATED WORK

**Offline-to-online RL.** Some offline RL algorithms are directly applied for O2O setting (Nair et al., 2020; Kumar et al., 2020; Kostrikov et al., 2022). A series of Q-ensemble based algorithms are proposed, while combined with balanced experience replay (Lee et al., 2022), state-dependent balance coefficient (Wang et al., 2024), uncertainty quantification guidance (Guo et al., 2023), uncertainty penalty and smoothness regularization (Wen et al., 2024) and optimistic exploration (Zhao et al., 2024). Model-based O2O RL algorithms combined with prioritized sampling scheme (Mao et al., 2022), or energy-guided diffusion sampling technique (Liu et al., 2024) are proposed to mitigate O2O distribution shift. Recently some work attempts to efficiently explore the environment to accelerate the fine-tuning process: O3F optimistically takes actions with higher expected Q-values (Mark et al., 2022), PEX introduces an extra policy to adaptively explore and learn (Zhang et al., 2023), OOO framework maintains an exploration policy to collect data and an exploitation policy to train on all data (Mark et al., 2024) and PTGOOD utilizes planning procedure to explore high-reward areas distant from offline distribution (McInroe et al., 2024). There are some other independent works: SPOT brings out a density-based regularization term to model the behavior policy (Wu et al., 2022), Td3+BC integrates behavioural cloning /constraint that decays over time (Beeson & Montana, 2022), Cal-QL calibrates the learned Q-values at reasonable scale with some reference policy (Nakamoto et al., 2023). OLLIE proposes the O2O imitation learning (Yue et al., 2024).

We observe that a current work WSRL explores to initialize the replay buffer without retaining offline data (Zhou et al., 2024). Nevertheless, it only adopts the Q-ensemble technique to resist

the distribution shift, while SAMG proposes an elegant and computationally efficient solution. Appendix D.7 offers a comprehensive comparison between SAMG and WSRL. We also note that DMG (Mao et al., 2024), an offline reinforcement learning algorithm, also involves mixing Q-values, but its core differs significantly from that of SAMG. Specifically, DMG mixes the maximum Q-values of in-distribution (ID) and OOD data and directly modifies the target values; in contrast, SAMG mixes offline and online Q-values and adaptively mixes the two target values. Furthermore, the two algorithms differ in both their algorithmic domains and theoretical frameworks.

## 7 CONCLUSION

This paper proposes a novel paradigm named SAMG to eliminate the tedious usage of offline data and leverage the pre-trained offline critic model instead, thereby ensuring 100% online sample utilization and better fine-tuning performance. SAMG seamlessly combines online and offline critics with a state-action-conditional coefficient without introducing undesirable or questionable intrinsic rewards. This coefficient estimates the complex distribution of the offline dataset and provides the probability of a given state-action sample. Theoretical analysis proves the convergence optimality and lower estimation error. Experimental results demonstrate the superiority of SAMG over vanilla baselines.

## ETHICS STATEMENT

This work focuses on offline-to-online reinforcement learning algorithms and does not involve human or animal subjects, personally identifiable data, or any interventions that could raise ethical concerns. No potentially harmful or offensive content is generated, and there are no safety issues associated with this research.

## REPRODUCIBILITY STATEMENT

We have made considerable efforts to ensure our work can be reproduced. The method is detailed in Sec. 3, including model design and training paradigm. Related environments and base algorithms are clearly stated in Sec. 5. The pesudo code is provided in Appendix D.1 and all training details are in Appendix D.2. These provide enough information to replicate our experiments. We also attach the source code in the supplementary materials.

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

# A    USE OF LARGE LANGUAGE MODELS (LLMS)

In preparing and writing this paper, we used large language models (LLMs) only as an auxiliary tool to help with language polishing and grammar checking. The research ideas, experimental design, analysis, and core technical writing were entirely carried out by the authors without relying on LLMs. The authors take full responsibility for the final cont ent.

# B    THEORETICAL ANALYSIS

## B.1    CONTRACTION PROPERTY

Our algorithm actually breaks the typical Bellman Equation of the RL algorithm denoted as $Q = \max_{\pi \in \Pi} \mathcal{B}^{\pi} = \max_{\pi \in \Pi}(r_{\pi} + \gamma P_{\pi}Q)$. Instead we promote Eq. (1). In order to prove the convergence of the updating equation, we introduce the contraction mapping theorem which is widely used to prove the convergence optimality of RL algorithm.

**Theorem B.1** (Contraction mapping theorem). *For an equation that has the form of $x = f(x)$ where $x$ and $f(x)$ are real vectors, if $f$ is a contraction mapping which means that $\|f(x_1) - f(x_2)\| \leq \gamma\|x_1 - x_2\|(0 < \gamma < 1)$, then the following properties hold.*

*Existence: There exists a fixed point $x^*$ that satisfies $f(x^*) = x^*$.*

*Uniqueness: The fixed point $x^*$ is unique.*

*Algorithm: Given any initial state $x_0$, consider the iterative process: $x_{k+1} = f(x_k)$, where $k = 0, 1, 2, ...$. Then $x_k$ convergences to $x^*$ as $k \to \infty$ at an exponential convergence rate.*

We just need to prove that this equation satisfies the contraction property of theorem B.1 and naturally we can ensure the convergence of the algorithm.

Take the right hand of Eq. (equation 1) as function $f(Q)$ and consider any two vectors $Q_1, Q_2 \in \mathbb{R}^{\mathcal{S}}$, and suppose that:

$$
\begin{aligned}
\pi_1^* &\doteq \arg\max_{\pi}(f(Q_1)) = \arg\max_{\pi}\left[(1 - p(s,a))\mathcal{B}^{\pi}Q_1 + p(s,a)\mathcal{B}^{\pi}Q^{off}\right] \\
\pi_2^* &\doteq \arg\max_{\pi}(f(Q_2)) = \arg\max_{\pi}\left[(1 - p(s,a))\mathcal{B}^{\pi}Q_2 + p(s,a)\mathcal{B}^{\pi}Q^{off}\right].
\end{aligned}
\tag{10}
$$

Then,

$$
\begin{aligned}
f(Q_1) &= \max_{\pi}\left[(1 - p(s,a))\mathcal{B}^{\pi}Q_1 + p(s,a)\mathcal{B}^{\pi}Q^{off}\right] \\
&= (1 - p(s,a))\mathcal{B}^{\pi_1^*}Q_1 + p(s,a)\mathcal{B}^{\pi_1^*}Q^{off} \\
&\geq (1 - p(s,a))\mathcal{B}^{\pi_2^*}Q_1 + p(s,a)\mathcal{B}^{\pi_2^*}Q^{off},
\end{aligned}
\tag{11}
$$

and similarly:

$$
f(Q_2) \geq (1 - p(s,a))\mathcal{B}^{\pi_1^*}Q_2 + p(s,a)\mathcal{B}^{\pi_1^*}Q^{off}.
\tag{12}
$$

To simplify the derivation process, we use $p^{\pi}$ to represent $p(s, \pi(a|s))$ considering that values of $p$ function class are determined by the policy $\pi$ of any given state. As a result,

$$
\begin{aligned}
&f(Q_1) - f(Q_2) \\
&= (1 - p^{\pi_1^*})\mathcal{B}^{\pi_1^*}Q_1 + p^{\pi_1^*}\mathcal{B}^{\pi_1^*}Q^{off} - \left[(1 - p^{\pi_2^*})\mathcal{B}^{\pi_2^*}Q_2 + p^{\pi_2^*}\mathcal{B}^{\pi_2^*}Q^{off}\right] \\
&\leq (1 - p^{\pi_1^*})\mathcal{B}^{\pi_1^*}Q_1 + p^{\pi_1^*}\mathcal{B}^{\pi_1^*}Q^{off} - \left[(1 - p^{\pi_1^*})\mathcal{B}^{\pi_1^*}Q_2 + p^{\pi_1^*}\mathcal{B}^{\pi_1^*}Q^{off}\right] \\
&= (1 - p^{\pi_1^*})(\mathcal{B}^{\pi_1^*}Q_1 - \mathcal{B}^{\pi_1^*}Q_2) \\
&= \gamma(1 - p^{\pi_1^*})P^{\pi_1^*}(Q_1 - Q_2) \\
&\leq \gamma P^{\pi_1^*}(Q_1 - Q_2).
\end{aligned}
\tag{13}
$$

We can see that the result reduces to that of the normal Bellman equation and therefore, the following derivation is omitted. As a result, we get,

$$\|f(Q_1) - f(Q_2)\|_\infty \le \gamma \|Q_1 - Q_2\|_\infty, \tag{14}$$

which concludes the proof of the contraction property of $f(Q)$.

## B.2 CONVERGENCE OPTIMALITY

We consider a tabular setting for simplicity. We first write down the iterative form of Eq. (6) as below:

if $s = s_k, a = a_k$,

$$Q_{k+1}(s,a) = Q_k(s,a) - \alpha_k(s,a)\big[Q_k(s,a) - (r_{k+1} + \gamma Q_k(s_{k+1}, a_{k+1}))\big]$$
$$- \alpha_k(s,a)\gamma p(s,a)\big(Q^{off}(s_{k+1}, a_{k+1}) - Q_k(s_{k+1}, a_{k+1})\big). \tag{15}$$

else,

$$Q_{k+1}(s,a) = Q_k(s,a). \tag{16}$$

The error of estimation is defined as:

$$\Delta_k(s,a) \doteq Q_k(s,a) - Q(s,\pi). \tag{17}$$

where $Q_\pi(s,a)$ is the state action value s under policy $\pi$. Deducting $Q_\pi(s,a)$ from both sides of 8 gets:

$$\Delta_{k+1}(s,a) = (1 - \alpha_k(s,a))\,\Delta_k(s,a) + \alpha_k(s,a)\eta_k(s,a), \quad s = s_k, a = a_k. \tag{18}$$

where

$$\eta_k(s,a)$$
$$= \big[r_{k+1} + \gamma Q_k(s_{k+1}, a_{k+1}) - Q^\pi(s,a)\big] + \gamma p(s,a)\big[Q^{off}(s_{k+1}, a_{k+1}) - Q_k(s_{k+1}, a_{k+1})\big]$$
$$= \big[r_{k+1} + \gamma Q_k(s_{k+1}, a_{k+1}) - Q^\pi(s,a)\big] +$$
$$\gamma p(s,a)\Big\{\big[Q^{off}(s_{k+1}, a_{k+1}) - Q^\pi(s_{k+1}, a_{k+1})\big] + \big[Q^\pi(s_{k+1}, a_{k+1}) - Q_k(s_{k+1}, a_{k+1})\big]\Big\}$$
$$= \underbrace{\big[r_{k+1} + \gamma Q_k(s_{k+1}, a_{k+1}) - Q^\pi(s,a)\big]}_{\Gamma_1} - \underbrace{\gamma p(s,a)\big[Q_k(s_{k+1}, a_{k+1}) - Q^\pi(s_{k+1}, a_{k+1})\big]}_{\Gamma_2}$$
$$+ \underbrace{\gamma p(s,a)\big[Q^{off}(s_{k+1}, a_{k+1}) - Q^\pi(s_{k+1}, a_{k+1})\big]}_{\Gamma_3}$$
$$= \Gamma_1 - \Gamma_2 + \Gamma_3. \tag{19}$$

Similarly, deducting $Q^\pi(s,a)$ from both side of Eq. (16) gets:

$$\Delta_{k+1}(s,a) = (1 - \alpha_k(s,a))\,\Delta_k(s,a) + \alpha_k(s,a)\eta_k(s,a), \quad s \ne s_k \text{ or } a \ne a_k.$$

this expression is the same as 18 except that $\alpha_k(s,a)$ and $\eta_k(s,a)$ is zero. Therefore we observe the following unified expression:

$$\Delta_{k+1}(s,a) = (1 - \alpha_k(s,a))\,\Delta_k(s,a) + \alpha_k(s,a)\eta_k(s,a).$$

To further analyze the convergence property, we introduce Dvoretzky's theorem (T. Jaakkola & Singh, 1994):

**Theorem B.2** (Dvoretzky's Throrem). *Consider a finite set $\mathcal{S}$ of real numbers. For the stochastic process:*

$$\Delta_{k+1}(s) = (1 - \alpha_k(s))\,\Delta_k(s) + \beta_k(s)\eta_k(s).$$

*it holds that $\Delta_k(s)$ convergences to zero almost surely for every $s \in \mathcal{S}$ if the following conditions are satisfied for $s \in \mathcal{S}$:*

*(a)* $\sum_k \alpha_k(s) = \infty$, $\sum_k \alpha_k^2(s) < \infty$, $\sum\limits_k \beta_k^2(s) < \infty$, $\mathbb{E}[\beta_k(s)|\mathcal{H}_k] \le \mathbb{E}[\alpha_k(s)|\mathcal{H}_k]$ *uniformly almost surely;*

*(b)* $\left\|\mathbb{E}[\eta_k(s)|\mathcal{H}_k]\right\|_\infty \le \gamma \left\|\Delta_k\right\|_\infty$, *with* $\gamma \in (0,1)$;

*(c)* $var\left[\eta_k(s)|\mathcal{H}_k\right] \le C\left(1 + \left\|\Delta_k(s)\right\|_\infty\right)^2$, *with* $C$ *a constant.*

*Here,* $\mathcal{H}_k = \{\Delta_k, \Delta_{k-1}, \cdots, \eta_{k-1}, \cdots, \alpha_{k-1}, \cdots, \beta_{k-1}, \cdots\}$ *denotes the historical information. The term* $\|\cdot\|_\infty$ *represents the maximum norm.*

To prove SAMG is well-converged, we just need to validate that the three conditions are satisfied. Nothing changes in our algorithm compared to normal RL algorithms when considering the first condition so it is naturally satisfied. Please refer to (T. Jaakkola & Singh, 1994) for detailed proof. For the second condition, due to the Markovian property, $\eta_t(s,a)$ does not depend on the historical information and is only dependent on $s$ and $a$. Then, we get $\mathbb{E}[\eta_k(s,a)|\mathcal{H}_k] = \mathbb{E}[\eta_k(s,a)]$.

Specifically, for $s = s_t, a = a_t$, we have:

$$\mathbb{E}[\eta_k(s,a)] = \mathbb{E}[\eta_k(s_k,a_k)] = \mathbb{E}[\Gamma_1] - \mathbb{E}[\Gamma_2] + \mathbb{E}[\Gamma_3].$$

For the first term,

$$\mathbb{E}[\Gamma_1] = \mathbb{E}\left[r_{k+1} + \gamma Q_k(s_{k+1}, a_{k+1}) - Q^\pi(s_k, a_k)\,|s_k, a_k\right]$$
$$= \mathbb{E}\left[r_{k+1} + \gamma Q_k(s_{k+1}, a_{k+1})\,|s_k, a_k\right] - Q^\pi(s_k, a_k).$$

Since $Q_\pi(s_k, a_k) = \mathbb{E}\left[r_{k+1} + \gamma Q^\pi(s_{k+1}, a_{k+1})\,|s_k, a_k\right]$, the above equation indicates that,

$$\mathbb{E}[\Gamma_1] = \gamma \mathbb{E}\left[Q_k(s_{k+1}, a_{k+1}) - Q^\pi(s_{k+1}, a_{k+1})\,|s_k, a_k\right].$$

For the second term,

$$\mathbb{E}[\Gamma_2] = \gamma p(s_k, a_k)\mathbb{E}\left[Q_k(s_{k+1}, a_{k+1}) - Q^\pi(s_{k+1}, a_{k+1})\,|s_k, a_k\right].$$

Combining these two terms gets:

$$\mathbb{E}[\Gamma_1] - \mathbb{E}[\Gamma_2] = \gamma\left(1 - p(s_k, a_k)\right)\mathbb{E}\left[Q_k(s_{k+1}, a_{k+1}) - Q^\pi(s_{k+1}, a_{k+1})\,|s_k, a_k\right].$$

Then,

$$\left\|\mathbb{E}[\Gamma_1] - \mathbb{E}[\Gamma_2]\right\|_\infty$$
$$= \gamma\left(1 - p(s_k, a_k)\right)\left\|\sum_{a' \in \mathcal{A}}\sum_{s' \in \mathcal{S}} t\left(s', a'\,|s_k, a_k\right)\left|Q_k(s', a') - Q^\pi(s', a')\right|\right\|_\infty$$
$$= \gamma\left(1 - p(s_k, a_k)\right)\max_{s' \in \mathcal{S}, a' \in \mathcal{A}}\left\{\sum_{a' \in \mathcal{A}}\sum_{s' \in \mathcal{S}} t\left(s', a'\,|s_k, a_k\right)\left|Q_k(s', a') - Q^\pi(s', a')\right|\right\}$$
$$\le \gamma\left(1 - p(s_k, a_k)\right)\sum_{a' \in \mathcal{A}}\sum_{s' \in \mathcal{S}} t\left(s', a'\,|s_k, a_k\right)\max_{s' \in \mathcal{S}, a' \in \mathcal{A}}\left[\left|Q_k(s', a') - Q^\pi(s', a')\right|\right]$$
$$= \gamma\left(1 - p(s_k, a_k)\right)\max_{s' \in \mathcal{S}, a' \in \mathcal{A}}\left[\left|Q_k(s', a') - Q^\pi(s', a')\right|\right]$$
$$= \gamma\left(1 - p(s_k, a_k)\right)\left\|\Delta_k(s, a)\right\|_\infty.$$

For the third term, to simplify the derivation, we mildly abuse the notation of $s', a'$ to represent $s_{k+1}, a_{k+1}$,

$$\mathbb{E}[\Gamma_3] = \mathbb{E}\left[\gamma p(s_k, a_k)\left(Q^{off}(s_{k+1}, a_{k+1}) - Q^\pi(s_{k+1}, a_{k+1})\right)\right]$$
$$= \gamma\sum_{a' \in \mathcal{A}}\sum_{s' \in \mathcal{S}} t\left(s', a'\,|s_k, a_k\right)p(s_k, a_k)\left|Q^{off}(s', a') - Q^\pi(s', a')\right|.$$

It follows that:

$$\left\|\mathbb{E}[\Gamma_3]\right\|_\infty$$

$$=\left\|\gamma \sum_{a'\in\mathcal{A}}\sum_{s'\in\mathcal{S}} t\left(s',a'\,|s_k,a_k\right)p(s_k,a_k)\left|Q^{off}(s',a')-Q^\pi(s',a')\right|\right\|_\infty$$

$$=\gamma p(s_k,a_k)\max_{s'\in\mathcal{S},a'\in\mathcal{A}}\left\{\sum_{a'\in\mathcal{A}}\sum_{s'\in\mathcal{S}} t\left(s',a'\,|s_k,a_k\right)\left|Q^{off}(s',a')-Q^\pi(s',a')\right|\right\}$$

$$\leq\gamma p(s_k,a_k)\sum_{a'\in\mathcal{A}}\sum_{s'\in\mathcal{S}} t\left(s',a'\,|s_k,a_k\right)\max_{s'\in\mathcal{S},a'\in\mathcal{A}}\left[\left|Q^{off}(s',a')-Q^\pi(s',a')\right|\right]$$

$$=\gamma p(s_k,a_k)\max_{s'\in\mathcal{S},a'\in\mathcal{A}}\left[\left|Q^{off}(s',a')-Q^\pi(s',a')\right|\right].$$

If the sample $(s',a')$ is in the distribution of offline dataset, We notice that the probability $(s',a')$ is significant and the $Q_k(s',a')$ is a good estimation of the optimal value $Q_\pi(s',a')$ and the specific form of TD error depends on the offline algorithm, and we can uniformly formulate this by:

$$\left\|\mathbb{E}[\Gamma_3]\right\|_\infty = \gamma\gamma_\mathcal{F}p(s_k,a_k)\max_{s',a'\in\mathcal{D}}\left[\left|Q^{off}(s',a')-Q^\pi(s',a')\right|\right]$$

$$= \gamma\gamma_\mathcal{F}p(s_k,a_k)\left\|\Delta_N(s,a)\right\|_\infty.$$

where $\mathcal{F}$ denotes the function class of offline algorithm, $0 < \gamma_\mathcal{F} < 1$ denotes the convergence coefficient of offline algorithm class $\mathcal{F}$ and $N$ denotes the iterative number of offline pre-training.

But while $(s',a')$ falls out of the distribution of offline dataset, the probability $p(s',a')$ is trivial with an upper bound constrained to a diminutive number $\xi_{OOD}$, denoted as $p(s',a') < \xi_{OOD}(s',a')$, and we know little about the $|Q_\pi(s',a')-Q_k(s',a')|$ but it is inherently restricted by the maximum reward $R_{max}$. Then this term is limited by $2\gamma\xi_{OOD}(s',a')R_{max}$ and we cut the probability $p(s',a')$ to zero in practice. Combining the above two cases gets the following upper limit:

$$\left\|\mathbb{E}[\Gamma_3]\right\|_\infty \leq \max\left\{\gamma\gamma_\mathcal{F}p(s_k,a_k)\left\|\Delta_N(s,a)\right\|_\infty, 2\gamma\xi_{OOD}(s',a')R_{max}\right\}$$

$$=\gamma\gamma_\mathcal{F}p(s_k,a_k)\left\|\Delta_N(s,a)\right\|_\infty.$$

Therefore,

$$\left\|\mathbb{E}[\eta_k(s,a)]\right\|_\infty =\left\|\mathbb{E}[\Gamma_1] - \mathbb{E}[\Gamma_2] + \mathbb{E}[\Gamma_3]\right\|_\infty$$

$$\leq\gamma\left(1 - p(s_k,a_k)\right)\left\|\Delta_k(s,a)\right\|_\infty + \gamma\gamma_\mathcal{F}p(s_k,a_k)\left\|\Delta_N(s,a)\right\|_\infty.$$

Because $N$ is big enough that $\left\|\Delta_N(s,a)\right\|_\infty$ is a high-order small quantity compared to $\left\|\Delta_k(s,a)\right\|_\infty$ and can be written as $\mathcal{O}(\left\|\Delta_k(s,a)\right\|_\infty)$. Therefore,

$$\left\|\mathbb{E}[\eta_k(s,a)]\right\|_\infty \leq\gamma\left(1 - p(s_k,a_k)\right)\left\|\Delta_k(s,a)\right\|_\infty + \gamma\gamma_\mathcal{F}p(s_k,a_k)\mathcal{O}(\left\|\Delta_k(s,a)\right\|_\infty). \tag{20}$$

where $0 < \gamma\left(1 - p(s_k,a_k)\right) < 1$ and the second condition is satisfied. Finally, regarding the third condition, we have when $s = s_k, a + a_k$,

$$var\left[\eta_k(s)|\mathcal{H}_k\right]$$

$$=var\left\{\left[r_{k+1} + \gamma Q_k(s_{k+1},a_{k+1}) - Q^\pi(s,a)\right] + \gamma p(s,a)\left[Q^{off}(s_{k+1},a_{k+1}) - Q_k(s_{k+1},a_{k+1})\right]\right\}.$$

and $var\left[\eta_k(s)|\mathcal{H}_k\right] = 0$ for $s \neq s_k$ or $a \neq a_k$.

Since $r_{k+1}$ and $\mathbb{E}\left[\eta_k(s)|\mathcal{H}_k\right]$ are both bounded, the third condition can be proven easily. And Therefore SAMG is well converged.

DISCUSSION OF THE CONVERGENCE

Theorem 4.1 establishes that the SAMG update rule is a convergent policy evaluation method. While this does not directly prove convergence to the optimal Q-function, $Q^*$, for the full Q-learning control

loop (which involves policy improvement), it is a critical prerequisite, demonstrating that the value estimation process itself is sound under our proposed modification.

While a full proof of convergence to $Q^*$ for SAMG in a control setting with function approximation is complex and beyond the scope of this paper's analysis, the convergence of the evaluation step is a positive indication of stability. SAMG operator only modifies and accelerates the policy evaluation process, and is integrated with algorithms like CQL, IQL and AWAC, which themselves have their own mechanisms for handling policy improvement.

While our current theoretical analysis focuses on the convergence of policy evaluation, the strong empirical performance of SAMG when integrated with established Q-learning based algorithms across various challenging benchmarks (Section 5) suggests its effectiveness in the practical control setting, leading to policies that achieve high returns.

### B.3 Convergence Speed

In this section, we give a more detailed analysis of the convergence speed of SAMG. For vanilla RL algorithms, the contraction coefficient $\gamma$ represents the convergence speed because it controls the contraction speed of Q-iteration. For SAMG, we give a rough derivation in Appendix B.1 that SAMG possesses a smaller contraction coefficient. But how small could that be? We actually have already derived the specific form of contraction factor in Appendix B.2, as specified in Eq. (20). However, Eq. (20) just covers the in-distribution situation of the contraction coefficient. As for the OOD situation, the contraction coefficient share the same coefficient as the normal Bellman equation. To sum up, we write the whole the contraction coefficient as below:

$$\gamma_o \leq \begin{cases} (1 - p(s_k, a_k))\, \gamma + \gamma \gamma_{\mathcal{F}} p(s_k, a_k) \dfrac{\left\| \Delta_{off}(s,a) \right\|_\infty}{\left\| \Delta_k(s,a) \right\|_\infty}, & p(s_k, a_k) \geq p_m \\ \gamma, & p(s_k, a_k) < p_m \end{cases} \tag{21}$$

where $0 < \gamma_{\mathcal{F}} < 1$ denotes the convergence coefficient of offline algorithm class $\mathcal{F}$ and $\left\| \Delta_{off}(s,a) \right\|_\infty$ denotes the offline suboptimality bound $[V^*(s) - V^{\pi_{off}}(s)]$ and $\left\| \Delta_k(s,a) \right\|_\infty$ denotes the suboptimality bound of the k-th iteration of online fine-tuning.

The upper equation holds for in-distribution samples, which are well mastered by the offline model. Therefore, the offline suboptimality bound is substantially tighter compared to the online bound. This illustrates that the offline model guidance significantly accelerates the online fine-tuning process by providing accurate estimations for in-distribution samples.

## C State-action-conditional coefficient

### C.1 Posterior collapse situation

we observe that the previous C-VAE structure suffers from posterior collapse, as shown in Fig. 3. Posterior collapse implies that the encoder structure completely fails. The KL-divergence loss vanishes to zero for any input, and the latent output is just a standard normal distribution (0 for the mean and 1 for the variance). Thus, the decoder structure takes noise $z \sim \mathcal{N}(0, 1)$ as input and reconstructs samples all by itself.

Fig. 3 illustrates the KL loss values with posterior collapse (with "s-hopper-m" and "s-hopper-me" legend) and without posterior collapse (with "sa-hopper-m" and "sa-hopper-me" legend), representing the distribution error between the output distribution of encoder and standard normal distribution. It can be observed that situations with posterior collapse possess much lower loss term (approximately by four orders of magnitude). Though the loss is lower for the posterior collapse situation, the output of all state-action samples are 0 and 1 for mean and standard respectively. Therefore the encoder totally fails to function.

### C.2 VAE implementations

For the C-VAE module, we employ the same VAE structure as Xu (Xu et al., 2022) except that we change the input to (state, action) and the output to next state. Furthermore, we adopt the KL-annealing technique in the hopper environments where we do not introduce the KL loss initially

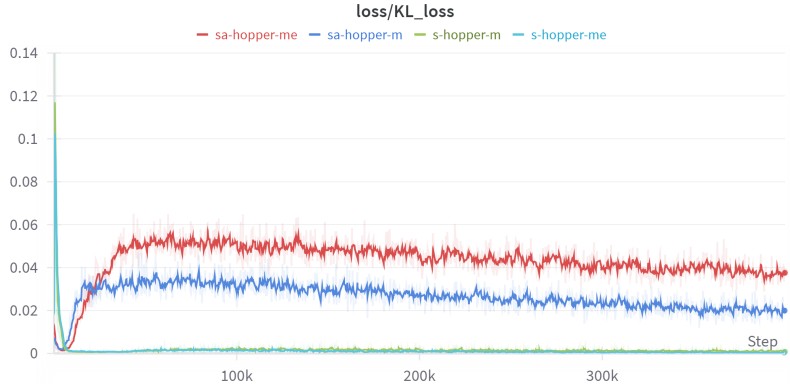

Figure 3: **Illustration of the posterior collapse of C-VAE structure** The blue curve represents the normal KL loss term while the green term represents the posterior collapse situation.

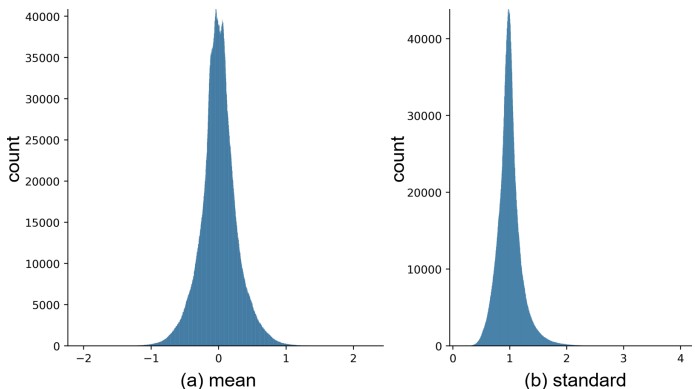

Figure 4: **Statistical results of the output from the C-VAE model, including (a) the mean values and (b) the standard values**

by manually setting it to zero and slowly increasing the KL loss weight with time. KL-annealing could result in more abundant representations of the encoder and is less likely to introduce posterior collapse. We also simplify the decoder of the C-VAE module in hopper and Walker2d environments to avoid posterior collapse. Notably, avoid normalizing the states and the actions because the normalized states are highly likely to result in the posterior collapse. In terms of experimental experience, the algorithm performs best when the KL loss converges to around 0.03. The information of the next state is supplemented in the training phase to better model the offline distribution and statistical techniques are combined with neural networks to obtain more reasonable probability estimation.

### C.3 PRACTICAL VAE DISTRIBUTION

In practice implementation, the offline data is input to C-VAE model, and the statistical result of output from the C-VAE model, including mean ($z_m$) and standard ($z_v$), are shown in the Fig. 4. From the figure we can conclude that the statistical distributions of the mean and standard exhibits a near-Gaussian distribution and fitting the variable with a normal distribution yields very small mean squared error: around 10 for the mean and 100 for the standard (with up to 1 million data). Therefore, we can consider the fitted distribution as the offline distribution and resort to this fitted distribution to infer the coefficient.

## C.4 DERIVATION OF THE PROBABILITY OF C-VAE

We derive the probability for the mean $Z_m$ below and the derivation for the standard is similar.

$$
\begin{aligned}
&P(|Z_m - \mu_m| > |z_m - \mu_m|) \\
=&P(Z_m > \mu_m + |z_m - \mu_m|) + P(Z_m < \mu_m - |z_m - \mu_m|) \\
=&2P(Z_m < \mu_m - |z_m - \mu_m|) \\
=&2F_{Z_m}(\mu_m - |z_m - \mu_m|)
\end{aligned}
\tag{22}
$$

where the second equation comes from the symmetry of the normal distribution about its mean and the second equation comes from the definition of the cumulative distribution function.

## C.5 ADAPTIVE VAE COEFFICIENT

At fixed intervals ($N_{update}$ steps, set to 10k in our implementation, meaning that only a few updates are required throughout the fine-tuning process), we first collect data from the current period (all data from the previous to the current interval) and then filter out OOD samples, whose $p^{off} < p_m^{off}$. From this set, we identify "mastered OOD samples". Since the model is lack of awareness of OOD samples, estimated Q-values tend to introduce significant errors and large online-critic loss terms during training. Therefore, the magnitude of the error between the estimated critic during online fine-tuning and the true Q-values can be used as a measure of how well the OOD samples have been mastered. However, in practice, the ground truth of the Q-values is unavailable, so the exact error can not be obtained. To address this, we adopted several potential approaches to estimate the error, which will be detailed later. After obtaining error estimates, we select samples with minimal errors as mastered OOD samples (set to less than 1e-1 in our implementation). These are OOD samples for which the online agent now demonstrates good predictive accuracy. These samples are then further fine-tune the existing VAE model parameters. This allows the VAE to expand its representation of "understood" or "in-distribution-like" state-action regions, consequently refining the coefficient for future online steps.

In the actual implementation, the error estimation methods replace the true Q-values with sampling (as referenced in (Kumar et al., 2020)) and use the practical Bellman operator, where the target Q-value serves as an estimate. We found that the results of these two methods are similar, with over 80% overlap in the filtered OOD samples, and both lead to comparable improvements in algorithm performance (with the sampling-based method performing slightly better). Considering the trade-off between the computation overhead and algorithm performance, we choose to use the Bellman operator. Additionally, we set the update interval to 10,000 steps. Since this interval exceeds the target Q network update interval (typically set to 1,000), we consistently refer to the target-Q network at the beginning of the period to ensure fairness. Given that the target network is already saved at this point, no additional computation overhead is introduced. Minimal error is defined as the smallest 10% of errors among the filtered samples.

Moreover, as the update of VAE model, the corresponding guidance of offline critic should also be updated as situations previously deemed out-of-distribution may now be well captured, with the associated probabilistic model having been revised accordingly. Consequently, the offline model can be substituted with the current Q-function model. This operation actually treats the current time step as the beginning of a new online phase, with all prior experiences regarded as offline knowledge. Since the algorithm relies on offline guidance during each update cycle, this transition does not introduce significant errors and retains the knowledge embedded in the previous offline model. Therefore, replacing the offline model with the current Q-function is a reasonable choice. Furthermore, continuously updating the offline model ensures that the algorithm progressively improves its understanding of the sample space, allowing the model to keep improving until convergence. We also experimented with the idea of adaptive tuning the threshold $p_m^{off}$. Specifically, after each update of the C-VAE, we collected C-VAE outputs on the offline dataset plus the set of mastered OOD samples, fitted the updated empirical distribution, and recomputed a new $p_m^{off}$ based on the distribution.

Empirically, we observed that the recomputed threshold changed only minimally throughout fine-tuning. This stability indicates that the initially estimated threshold is already a good approximation and that a fixed threshold works reliably in practice. Therefore, we opted for using a fixed value,

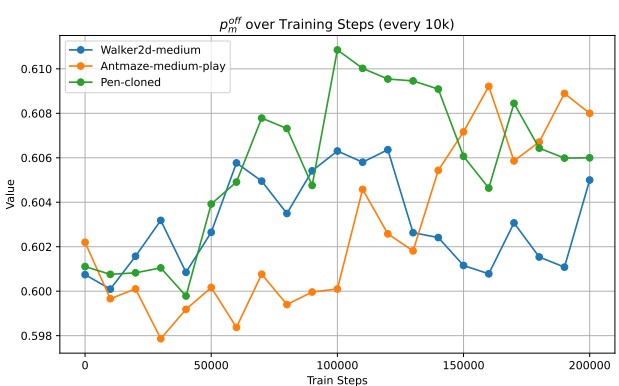

Figure 5: The values of $p_m^{off}$ through training.

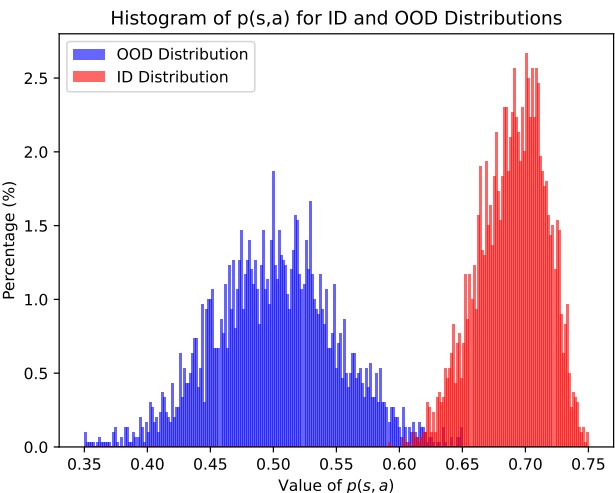

Figure 6: The histogram of $p(s, a)$ of ID and OOD dataset, the results are average across the Mujuco locomotion tasks. The ordinate represents the data percentage of each histogram bin.

which simplifies the algorithm, avoids additional computation overhead, and still provides accurate and stable performance. The values of $p_m^{off}$ through training is shown below:

## C.6 THE RELIABILITY OF C-VAE MODULE

To illustrate the reliability of C-VAE module as the state-action-conditional coefficient, we conducted the following two experiments. First, after pre-training the VAE, we collected two datasets: in-distribution (ID) and out-of-distribution (OOD), to showcase the model's excellent ability to distinguish data from different sources. Specifically, the OOD dataset was selected by choosing samples with the lowest trajectory similarity to the ID dataset across all levels of the environment. We present the output statistics of $p(s, a)$ on both datasets (as shown in Fig. 6), and the results indicate that the VAE exhibits strong capability in distinguishing ID from OOD samples. Additionally, this discriminative ability provides effective guidance for hyperparameter selection—setting it to approximately 0.6 yields optimal results.

To further visualize the modeling capability of the VAE, we analyzed the latent space outputs of the C-VAE for both datasets and applied t-SNE dimensionality reduction to project these outputs into a 2D space, as shown in Fig 7. In the figure, light blue points represent in-distribution (ID) samples, while dark blue points represent out-of-distribution (OOD) samples. The results indicate high overlap between ID and OOD samples in the t-SNE-reduced space, with only a few discrete OOD points.

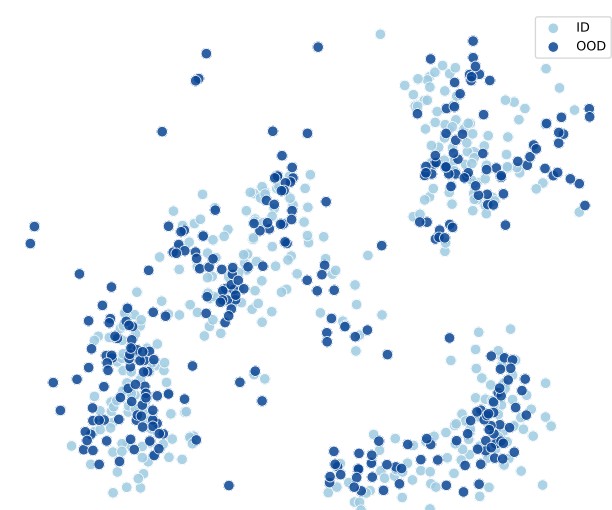

Figure 7: Visualization results of the latent space output by the VAE, dimensionality-reduced to 2D using t-SNE. Here, light blue points represent in-distribution samples, while dark blue points denote out-of-distribution samples.

This demonstrates that the C-VAE can effectively model the state-action density and cover both ID and OOD samples, further verifying its excellent modeling ability.

We also present a specific example. For instance, in Halfcheetah, the joint acceleration at the start of the environment is generally positive (indicating forward acceleration applied to the joints), which yields positive rewards—most samples in the offline dataset fall into this category. In contrast, we examined samples with negative joint acceleration at the environment's initiation, which are completely out-of-distribution. We found that the C-VAE outputs approximately 0.7 for the former and 0.4 for the latter, demonstrating its excellent modeling capability.

## C.7   HARD CUTOFF V.S. SOFT SCHEDULE FOR $p(s, a)$

We experimented with soft schedules instead of hard cutoff, denoted as SAMG-S, which performed slightly worse in our standard benchmarks, as shown in Table 3. The hard cutoff simplifies implementation and ensures that OOD samples with very low probability do not dominate updates.

Table 3: Hard cutoff v.s. Soft Schedule for $p(s, a)$

|          | Hopp-mr | Hopp-m | Ant-md | Ant-mp |
|----------|---------|--------|--------|--------|
| IQL      | 86.2    | 62.1   | 76.4   | 76.2   |
| SAMG     | 100.4   | 68.4   | 96.6   | 95.2   |
| SAMG-S   | 98.0    | 67.4   | 94.8   | 92.0   |

## C.8   CURVES OF $p(s, a)$ DURING ONLINE FINE-TUNING

We plotted and present the training curves of $p(s, a)$, focusing on two representative environments: door-cloned and halfcheetah-medium-replay. The results show that $p(s, a)$ gradually decreases with training, indicating the agent transitions from in-distribution (ID) samples to out-of-distribution (OOD) samples—this behavior is fully consistent with expectations.

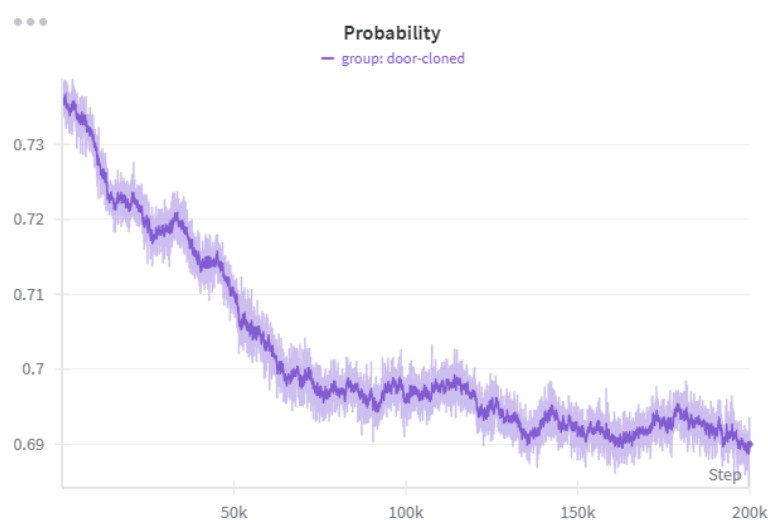

Figure 8: The curves of the state-action-conditional coefficient through the training on environment Door-cloned.

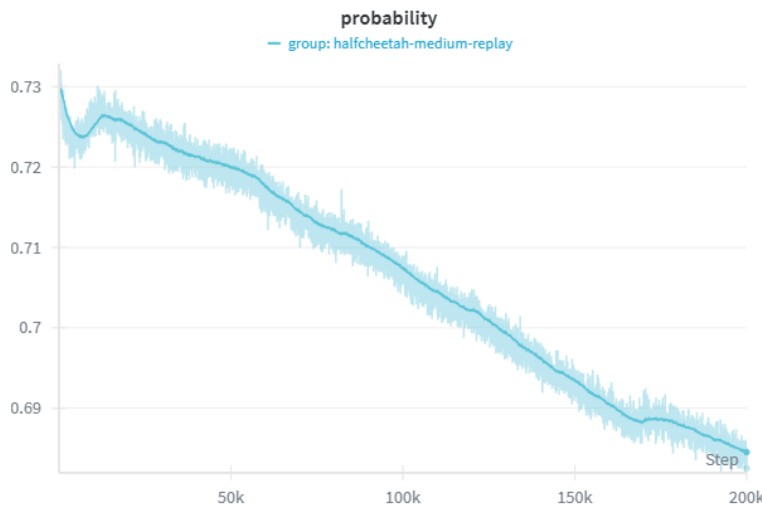

Figure 9: The curves of the state-action-conditional coefficient through the training on environment Halfcheetah-medium-replay.

## D    ALGORITHM IMPLEMENTATION

### D.1    SAMG PSEUDO-CODE

To illustrate the whole procedure of SAMG, we represent the pseudo-code of SAMG implemented based on AWAC (Nair et al., 2020) below:

### D.2    IMPLEMENTATION DETAILS

Only some minimal adjustments are needed to implement SAMG on AWAC, and IQL (Kostrikov et al., 2022) as well. We just need to maintain a much smaller replay buffer filled with online samples and insert and sample from this "online replay buffer". Before conducting normal gradient update step, we need to calculate the mixed $Q_{target}$ according to Eq. (6). As for IQL, we freeze and query

---

**Algorithm 1** Offline-to-Online Reinforcement Learning via State-Action-Conditional Offline Model Guidance (Implemented on AWAC)

---

**Require:** offline Q-network $Q_\phi^{off}$, policy $\pi_\theta^{off}$ and trained VAE model

1: $\pi_\theta \leftarrow \pi_\theta^{off}$, $Q_\phi \leftarrow Q_\phi^{off}$
2: Initialize the replay buffer $D$ with $N$ samples collected by $Q_\phi$
3: **for** iteration $i = 1, 2, ...$ **do**
4:     **for** every environment step **do**
5:         $a_t \sim \pi_\phi(s_t|s_t)$
6:         $s_{t+1}, d_t \sim p(s_{t+1}|s_t, a_t)$
7:         insert $(s_t, a_t, r_t, s_{t+1}, d_t)$ into D
8:     **end for**
9:     **for** every update step **do**
10:         get $Q_{target}$ and $Q_{target}^{off}$ according to AWAC
11:         get $p_m^{off}$ with C-VAE according to Section 3.2
12:         $Q_{target} \leftarrow (1 - p_{om})Q_{target} + p_{om}Q_{target}^{off}$
13:         Update $\phi$ according to Eq. 9 in (Nair et al., 2020) with $Q_{target}$
14:         Update $\theta$ according to Eq. 13 in (Nair et al., 2020)
15:         **if** This step % coefficient update interval == 0 **then**
16:             Filter mastered OOD samples
17:             Finetune the VAE model with collected samples
18:             Update the VAE model and the offline critic
19:         **end if**
20:     **end for**
21: **end for**

---

the offline pre-trained value function instead because IQL separately trains a value function to serve as the target information. Other implementations are similar to AWAC and are omitted.

As for CQL and Cal_QL, these two algorithms share a similar implementation procedure and align with AWAC when calculating the $Q_{target}$. However, CQL adds one extra penalty term to minimize the expected Q-value based on a distribution $\mu(a|s)$, formulated as $\mathbb{E}_{s\sim\mathcal{D}, a\sim\mu(a|s)}[Q(s,a)]$. This term is separated from the standard Bellman equation and serves an important role in making sure the learned Q-function is lower-bounded. However, this term is unrestricted in our paradigm and may cause algorithm divergence. So we add an offline version of the term still weighted by the state-action-conditional coefficient. This slightly avoids our setting but is reasonable that this setting shares the consistent updating direction with the Bellman equation error term.

We implement all the algorithms based on the benchmark CORL (Tarasov et al., 2024), whose open source code is available at `https://github.com/tinkoff-ai/CORL` and the license is Apache License 2.0 with detail in the GitHub link. Our code is attached in the supplementary material.

In practice, we strictly adopt the CORL setting to train the offline model and the vanilla fine-tuning training, including the training process and hyperparameters. As for SAMG training, for mujoco environments (halfcheetah, hopper, walker2d), SAMG algorithms share the same set of hyperparameters with the fine-tuning process to illustrate fairness. In the antmaze environment, we slightly reduce the weight of the Q-value maximization term, which corresponds to the $\alpha$ hyperparameter of CQL, to highlight the impact of SAMG for algorithms CQL and Cal_QL, from 5 to 2. For the threshold $p_m^{off}$, we adopt the value of 0.6 in most environments (including HalfCheetah, Hopper, Walker2d and Adroit) which seems large but only a small portion satisfies the condition. For the antmaze environment, we take 0.7 for CQL and 0.6 for the others. We found that in our setting, reducing the size of the replay buffer allows for more efficient utilization of samples, thereby improving the algorithm's performance. Specifically, we set the buffer size to be 50,000 for Anrmaze environment and 20,000 for the other environments. We initialize the replay buffer with 2000 samples utilizing the offline model (2000 is the normal length of an episode in most environments). The details of C-VAE have been stated in Appendix C. All the hyperparameters are summarized below:

Table 4: Hyperparameters for AWAC and IQL.

|  | Locomotion | Antmaze | Door | Pen |
|---|---|---|---|---|
| learning rate of SAMG | 1e-4 | 2e-5 | 1e-4 | 1e-4 |
| $p_m^{off}$ | 0.6 | 0.6 | 0.6 | 0.6 |
| Update frequency | 10k | 10k | 10k | 10k |
| Size of replay buffer | 20k | 50k | 20k | 20k |
| Latent dimension of C-VAE | 256 | 512 | 256 | 256 |
| learning rate of C-VAE | 1e-3 | 1e-3 | 1e-3 | 1e-3 |
| batch size | 32 | 32 | 32 | 32 |

Table 5: Hyperparameters for CQL.

|  | Locomotion | Antmaze | Door | Pen |
|---|---|---|---|---|
| learning rate of SAMG | 1e-4 | 2e-5 | 1e-4 | 1e-4 |
| $p_m^{off}$ | 0.6 | 0.7 | 0.6 | 0.6 |
| Update frequency | 10k | 10k | 10k | 10k |
| Size of replay buffer | 20k | 50k | 20k | 20k |
| Latent dimension of C-VAE | 256 | 512 | 256 | 256 |
| learning rate of C-VAE | 1e-3 | 1e-3 | 1e-3 | 1e-3 |
| batch size | 32 | 32 | 32 | 32 |

### D.3 DATASETS

D4RL (Datasets for Deep Data-Driven Reinforcement Learning) (Fu et al., 2020) is a standard benchmark including a variety of environments. SAMG is tested across four environments within D4RL: HalfCheetah, hopper, walker2d and antmaze.

1 **HalfCheetah**: The halfcheetah environments simulates a two-legged robot similar to a cheetah, but only with the lower half of the cheetah. The goal is to navigate and move forward by coordinating the movements of its two legs. It is a challenging environment due to the complex dynamics of the motivation.

2 **Hopper**: In the Hopper environment, the agent is required to control a one-legged hopping robot, whose objective is similar to that of the HalfCheetah. The agent needs to learn to make the hopper move forward while maintaining balance and stability. The Hopper environment presents challenges related to balancing and controlling the hopping motion.

3 **Walker2d**: Walker2d is an environment controlling a two-legged robot, which resembles a simplified human walker. The goal of Walker2d is to move the walker forward while maintaining stability. walker2d poses challenges similar to HalfCheetah environment but introduces additional complexities related to humanoid structure. The above three environments have three different levels of datasets, including medium-expert, medium-replay, medium.

4 **AntMaze**: In the AntMaze environment, the agent controls an ant-like robot to navigate through maze-like environments to reach a goal location. The agent receives a sparse reward that the agent only receives a positive reward when it successfully reaches the goal. this makes the task more difficult. The maze configurations vary from the following environments that possess different level of complexity, featuring dead ends and obstacles. There are totally six different levels of datasets, including: maze2d-umaze, maze2d-umaze-diverse, maze2d-medium-play, maze2d-medium-diverse, maze2d-large-play, maze2d-large-diverse.

5 **Adroit**: In the Adroit environment, the agent controls a robotic hand to finish various manipulation tasks, including pen balancing, door opening and object relocation. The agent only receives a positive reward when the task is successfully completed, otherwise, the reward is zero, which makes the tasks more challenging. We focus on three specific tasks: the **pen** agent must manipulate a pen to keep it balanced in some orientation; the **door** agent must grasp and open a door handle; the **relocate** agent must pick up an object and move it to a target location. We adopt the mixed setting of Cal_QL (Nakamoto et al., 2023), which combines the cloned-level and human-level dataset. However, we follow the reward mechanism used in the "cloned" environment, rather than adopting the binary reward formulation used in Cal_QL, which can explain the negative reward in these three

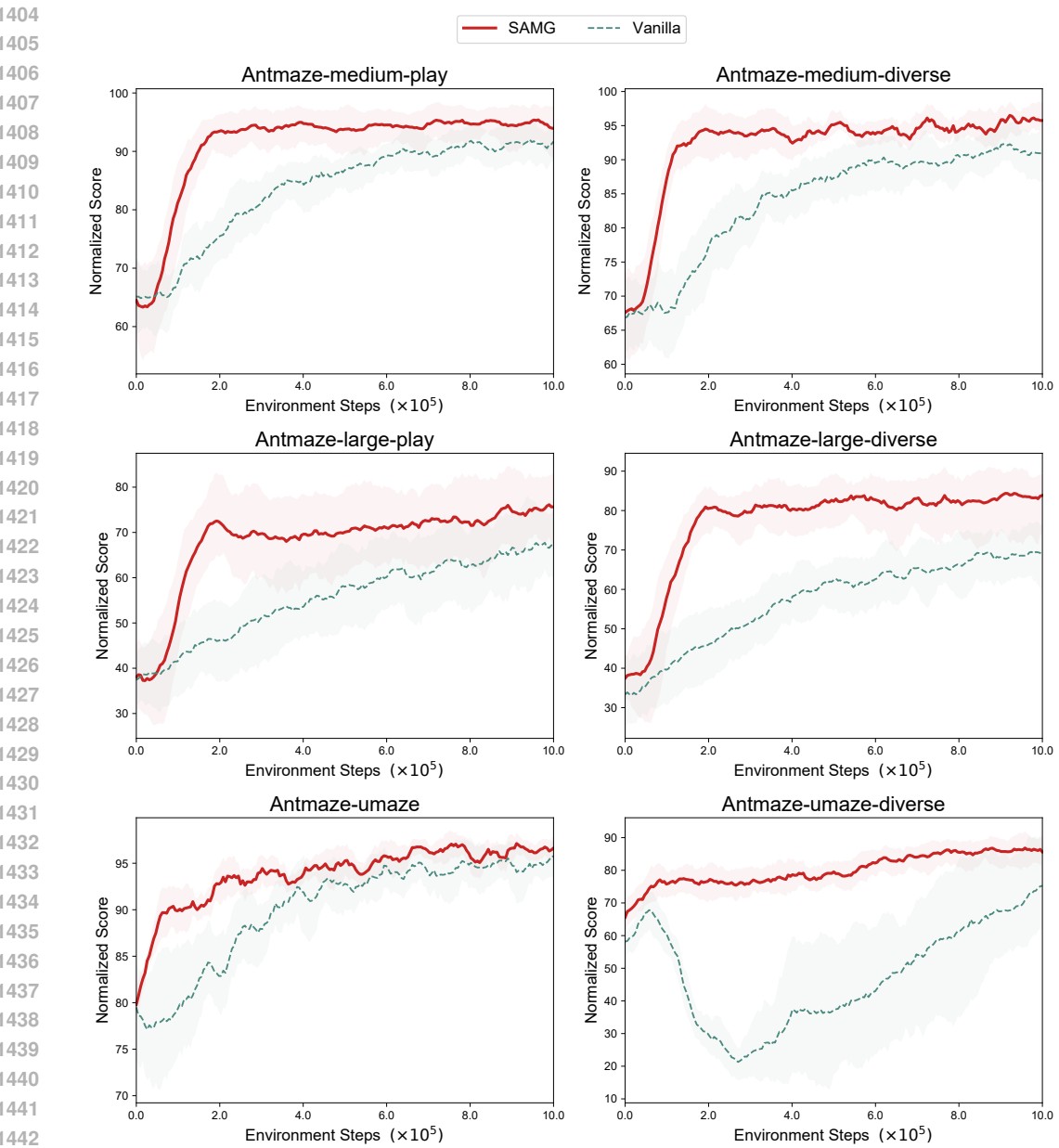

Figure 10: **Performance comparison**. This figure illustrates the asymptotic performance of IQL-based SAMG (denoted as SAMG) and IQL (denoted as Vanilla) across Antmaze tasks with 5 random seeds

environments: the reward is normalized between a random policy and an expert policy. As a result, algorithms that perform worse than the random policy achieve negative rewards—a phenomenon commonly observed in the Adroit environment (see CORL (Tarasov et al., 2024) for reference). We also denote them as "cloned" and thus introduces tree different tasks: Pen-cloned, Door-cloned and Relocate-cloned.

## D.4 SAMG PERFORMANCE

We present the training curves of IQL and SAMG (based on IQL) in Fig. 10. All experiments are conducted with five random seeds, and the results show that SAMG converges much faster than IQL while achieving a better final convergence value.

Table 6: **Cumulative regret of online fine-tuning algorithms.** The cumulative regret of standard base algorithms (including CQL, IQL, AWAC and Cal_QL, denoted as "Vanilla") compared to SAMG integrated algorithms (referred to as "Ours"). The result is the average normalized score of 5 random seeds $\pm$ (standard deviation). All algorithms are conducted for 500k iterations.

| Dataset | CQL | | AWAC | | IQL | | Cal_QL | |
|---|---|---|---|---|---|---|---|---|
| | Vanilla | Ours | Vanilla | Ours | Vanilla | Ours | Vanilla | Ours |
| antmaze-u | $0.051_{(0.005)}$ | $0.021_{(0.002)}$ | $0.081_{(0.046)}$ | $0.080_{(0.021)}$ | $0.072_{(0.005)}$ | $0.063_{(2.0)}$ | $0.023_{(0.003)}$ | $0.031_{(0.002)}$ |
| antmaze-ud | $0.185_{(0.061)}$ | $0.191_{(0.075)}$ | $0.875_{(0.046)}$ | $0.378_{(0.090)}$ | $0.392_{(0.116)}$ | $0.182_{(0.021)}$ | $0.142_{(0.124)}$ | $0.133_{(0.091)}$ |
| antmaze-md | $0.148_{(0.004)}$ | $0.131_{(0.010)}$ | $1.0_{(0.0)}$ | $1.0_{(0.0)}$ | $0.108_{(0.007)}$ | $0.102_{(0.008)}$ | $0.069_{(0.012)}$ | $0.076_{(0.025)}$ |
| antmaze-mp | $0.136_{(0.023)}$ | $0.078_{(0.369)}$ | $1.0_{(0.0)}$ | $1.0_{(0.0)}$ | $0.115_{(0.009)}$ | $0.143_{(0.020)}$ | $0.057_{(0.009)}$ | $0.071_{(0.008)}$ |
| antmaze-ld | $0.359_{(0.036)}$ | $0.382_{(0.023)}$ | $1.0_{(0.0)}$ | $1.0_{(0.0)}$ | $0.367_{(0.033)}$ | $0.305_{(0.041)}$ | $0.223_{(0.111)}$ | $0.219_{(0.157)}$ |
| antmaze-lp | $0.344_{(0.023)}$ | $0.317_{(0.052)}$ | $1.0_{(0.0)}$ | $1.0_{(0.0)}$ | $0.335_{(0.032)}$ | $0.321_{(0.043)}$ | $0.203_{(0.095)}$ | $0.211_{(0.114)}$ |

As illustrated in Section 5.1, SAMG performs best when integrated with AWAC compared to other algorithms.

The reason why AWAC-SAMG performs the best is detailed below. AWAC stands for advantage weighted actor critic, which is an algorithm to optimize the advantage function $A^{\pi_k}(s, a)$, while constraining the policy to stay close to offline data. AWAC does not contain any other tricks to under-estimate the value function as other offline RL algorithms (Kumar et al., 2020; Nakamoto et al., 2023), therefore AWAC could produce an accurate estimation of the values of offline data and serves as a perfect partner of SAMG.

For the other algorithms, they adopt various techniques to achieve conservative estimation of $Q$ values in order to counteract the potential negative effects of OOD samples. Therefore, the offline guidance they provide is a little less accurate. However, these algorithms are more robust due to conservative settings and can cope with more complex tasks, as illustrated in Section 5 of CQL (Kumar et al., 2020). However, it is always impossible to produce ideal $Q$ values for offline RL algorithms due to the limitations of offline datasets. The offline models trained by these algorithms could still provide guidance for the online fine-tuning process because the error of the estimation is trivial and the guidance is valuable and reliable. Furthermore, to resist the negative impact of conservative estimation, we cut the offline guidance and revert to the vanilla algorithms after a specific period of time in practice.

Overall, SAMG is a novel and effective paradigm, which is coherently conformed by theoretical analysis and abundant experiments.

## D.5 CUMULATIVE REGRET

The cumulative regrets of the Antmaze environment of four vanilla algorithms and SAMG are shown in Table 6.

It can be concluded from the table that SAMG possesses significantly lower regret than the vanilla algorithms, at least 40.12% of the vanilla algorithms in the scale. This illustrate the effectiveness of our algorithms in utilizing online samples and experimentally demonstrates the superiority of SAMG paradigm.

## D.6 UNSATISFACTORY PERFORMANCE ON PARTICULAR ENVIRONMENT

We think the hyperparameter $\tau$ and the environment property may account for the unsatisfactory performance of IQL-SAMG on the environment Halfcheetah-medium-expert, rather than the integration of IQL and SAMG.

In detail, as stated in the paper on IQL, the estimated function will gain on the optimal value as $\tau \to 1$. However, $\tau$ is not chosen to be 1 in practice and is quite low in the poorly performing environment. Additionally, this environment is relatively narrow and the training score is abnormally higher than the evaluation score. Therefore, we believe the unsatisfactory performance in this environment is just an exception and does not indicate problems of the SAMG paradigm.

### D.7 COMPARISON WITH WSRL

WSRL (Zhou et al., 2024) identifies the issue of not retaining the offline dataset, which is highly valuable. However, offline data can resist visiting too many online data, thus resisting the impact of distribution shift. However, directly discarding the offline data on top of previous O2O RL algorithms can lead to severe degradation, as the distribution shift is much severe. Therefore, additional mechanisms are required to mitigate the impact.

WSRL takes a relatively straightforward approach to compensate for the absence of offline data, including warm-up and Q-ensemble techniques. As for the method, rather than simply utilizing more models, we take a deeper focus on the model itself, effectively leveraging its information to resist distribution shift without introducing excessive complexity. As for the warm up, we also adopt a warm-up strategy as discussed in Appendix D.1. Both warm-up strategies are reasonable, which is also confirmed by WSRL in Appendix L in (Zhou et al., 2024).

In certain environments, SAMG (implemented based on CQL) achieves performance comparable to WSRL, as shown in Table 7.

Table 7: **Algorithm performance of WSRL and SAMG.** The algorithms performance of WSRL compared to SAMG(implemented based on CQL). The result is the average normalized score of 5 random seeds. The notions of each environments are the same as Table 1.

|      | Hopp-mr | Hopp-m | Hopp-me | Half-mr | Half-m | Half-me | Walk-mr | Walk-m | Walk-me |
|------|---------|--------|---------|---------|--------|---------|---------|--------|---------|
| SAMG | 103.7   | 88.3   | 113.0   | 57.8    | 59.0   | 97.2    | 88.4    | 82.9   | 112.5   |
| WSRL | 69.3    | 73.8   | 96.2    | 78.4    | 83.5   | 102.4   | 101.0   | 81.2   | 85.1    |

Furthermore, SAMG is also compatible with Q-ensemble technique, and combining the two yields significant performance improvements, which are substantially superior to the performance of WSRL, as shown in Table 8.

Table 8: **Algorithm performance of WSRL, SAMG and Q-ensemble-based SAMG.**

|               | Ant-ld | Ant-lp | Ant-mp | Ant-md |
|---------------|--------|--------|--------|--------|
| SAMG          | 63.8   | 60.8   | 86.4   | 89.2   |
| WSRL          | 90.0   | 87.6   | 90.0   | 85.0   |
| SAMG-ensemble | 95.0   | 96.4   | 100.0  | 96.0   |

As for the computational burdens of SAMG and WSRL, assuming the same base model is used (typically ranging from 1MB for SAC based algorithm to 10MB, denoted as $C$), SAMG only requires computational resources proportional to $2C + c$, where $c \approx 70kB$ is the size of the VAE model under our architecture. In contrast, WSRL requires approximately $10C$ compute, which demonstrates a substantially higher computational burden than SAMG.

### D.8 DATA COVERAGE RATE OF OFFLINE DATASET

To get the data coverage of a specific dataset, we aggregate all levels of datasets of a given environment, i.e., expert, medium-expert, medium-replay, medium, random level of datasets of environments HalfCheetah, Hopper, Walker2d. Thinking that the state and action are high-dimensional, we first perform dimensionality reduction. We uniformly and randomly select part of the data due to its huge scale and then perform t-SNE (Van der Maaten & Hinton, 2008) separately on the actions and states of this subset for dimensionality reduction. Given that it is hard to model the distribution of the continuous dimensional-reduced data, We then conduct hierarchical clustering (Müllner, 2011) to calculate and analyze the distribution of the data. We compute the clustering results of each environment and calculate the coverage rate based on the clustering results. To be specific, we select 10 percent of all data each time to cluster and repeat this process for 10 random seeds. For each clustering result, we calculate the data coverage rate of each level of offline dataset by counting the proportion of clustering center points. We consider one level of offline dataset to possess a clustering center if there exist more than 50 samples labeled with this clustering center.

# E LIMITATIONS

The performance improvement is limited if the offline dataset distribution is extremely narrow. This limitation could potentially be mitigated by designing specific update strategies for OOD samples, which is an interesting direction for future work.

# F COMPUTE RESOURCES

All the experiments in this paper are conducted on a Linux server with Intel(R) Xeon(R) Gold 6226R CPU @ 2.90GHz and NVIDIA Geforce RTX 3090. We totally use 8 GPU in the experiments and each experiment takes one GPU and roughly occupies around 30% of the GPU. It takes approximately 3 hours to 24 hours to run an experiment on one random seed, depending on the specific algorithms and environments. Specifically, the average time cost of experiments on Mujoco environments (HalfCheetah, Hopper, Walker2d) is 4.5 hours while it takes an average time of 20 hours in environment AntMaze. All experiments took a total of two months. Approximately ten days were spent on exploration, while twenty days were dedicated to completing preliminary offline algorithms.

# G POTENTIAL SOCIETAL IMPACTS

Our paradigm SAMG could be plugged in a variety of O2O RL algorithms and implemented with a small amount of computational cost. SAMG share similar societal impacts most offline-to-online RL algorithms. SAMG could not guarantee that the performance will always improve in the online fine-tuning process and the performance may fulctuate, which is limited by the offline RL setting. Furthermore,SAMG could not promise 100 percent safe decision-making, which aligns with most RL algorithms. Therefore, we suggest that the SAMG users should notice the potential risks and cautiously and safely use SAMG in online environments.

