# OpenReview forum: "SAMG: Offline-to-Online Reinforcement Learning via State-Action-Conditional Offline Model Guidance"
_ICLR.cc/2026/Conference — Submitted to ICLR 2026_

### Official Review · Reviewer_vCJH · 2025-10-30

**Soundness:** 2
**Presentation:** 3
**Contribution:** 3
**Rating:** 4
**Confidence:** 3

**Summary:**

This paper proposes a novel offline-to-online (O2O) reinforcement learning algorithm called SAMG that eliminates the need for offline data to be consumed again for online fine-tuning via state-action-conditional model guidance. SAMG first trains the critic and generative conditional model (C-VAE) in an offline manner, followed by online fine-tuning where the offline critic is frozen and used for guiding the online critic weighted by the state-action-conditional coefficient. The authors provide a theoretical analysis to prove the convergence property of SAMG with the modified Bellman operator, supported by empirical experiments with SOTA baselines on the D4RL benchmark. Comparisons with vanilla methods without SAMG highlight the clear improvements of the suggested online fine-tuning approach without the offline data.

**Strengths:**

- The idea of weighting the offline and online target critics with a confidence coefficient is simple but effective, which is reusable across most value-based offline RL methods.
- The contribution is strengthened by highlighting the difference with WSRL, which emphasizes the importance of eliminating the consumed offline data.
- Extensive empirical margins of adopting SAMG to popular offline RL methods demonstrate strong plug-and-play capability.
- Theoretical analysis helps in understanding the modified Bellman operator with the state-action-conditional coefficient, which serves the core function in the main methodology.

**Weaknesses:**

- The novelty of SAMG is marginal. Confidence coefficient estimation relies on the offline generative model for judging whether a given state-action pair is in-distribution or out-of-distributional data. However, there is no guarantee that the generative model produces reliable predictions when out-of-distributional data whether the output is erroneous or not. Furthermore, the fact that distinguishing whether the given pair falls under the offline distribution depends on the task-specific, hand-engineered threshold value $p_m^\text{off}$ exacerbates the reliability problem, since it is hard to decouple between erroneous output from the C-VAE or an inappropriate threshold value when the coefficient predicts the given pair is out-of-distributional data. I believe that providing further justifications on how well SAMG can overcome when C-VAE produces unreliable predictions or authors' tips on controlling the threshold value would strengthen the novelty.
- Experimental rigor could be improved significantly.
- - In Section 5.2, lacks of the experimental setup weaken the contribution of this paper. What does "1/2 for offline and 1/2 for online in SAMG (even)" exactly stand for? How do you generate the random probability for SAMG-random? What is the dataset used for this section (I infer it is *medium-replay* by comparing the results in Table 1)
- - In L205, the authors denote that samples with $p^\text{int} < p_m^\text{off}$ are regarded as OOD. However, the authors remark that if $p_m^\text{off}$ is too small, all samples will be regarded as OOD samples, thereby the online target critic is solely updated with the offline target critic (L419). This contradicts Equation (5) where $p^\text{off}(s,a)=0$ (OOD situation) when $p^\text{int}(s,a)<p_m^\text{off}$, since extremely low $p_m^\text{off}$ value would naviely pass most $p^\text{int}(s,a)$ into $p^\text{off}(s,a)$ directly. If this extreme case does not fall into an exemplary case, I would appreciate further justifications.
- Figure 1 can be improved to provide a comprehensive view of the overall architecture. I find it hard to establish an intuition about static or adaptive coefficient updates in Section 3.3.

**Questions:**

- What does $\alpha$ stand for in L257? The notation of the state-action-conditional coefficient is $p(s,a)$ or $p^\text{off}$, which is different.
- It is hard to find the best algorithms across different datasets and tasks in Table 1. Could you highlight the best scores per case in the table? (e.g., bold face)
- Although the authors note that the learning curves for $p_m^\text{off}=0.2$ and $p_m^\text{off}=0.5$ cases are overlapped in L421, readers may become confused when first seeing the plot alone. Could you provide more details in the caption in Figure 3?
- Extra experiments could deliver a more pronounced contribution of SAMG. I suggest a few recommendations below:
- - Curves of $p^\text{int}(s,a)$ (or $p^\text{off}(s,a)$ during online fine-tuning across different dataset optimalities. Ideally, the offline agent should rely more on the online target critic progressively as the fine-tuning progresses and the agent acquires better knowledge of the underexplored state-action space. I wonder how the coefficient degrades as the fine-tuning process continues.
- - If it is possible to attach a PDF file during the rebuttal, t-SNE visualization of latent vectors from C-VAE under two scenarios-in-distribution and out-of-distribution data is given-would justify the above weakness 1 by showing the similar clusters regardless of the distribution.

---

> ### Author Response · Authors · 2025-11-25
>
> The reviewer acknowledges the effectiveness of confidence-weighted critics, the clear distinction from WSRL, the strong plug-and-play empirical gains, and the supportive theoretical analysis. We respond to the detailed comments point by point below.
>
> >**Weakness 1 & Question 4.2: Reliability of VAE on OOD Samples**
>
> Thank you for the insightful comment. We clarify the novelty concerns regarding the coefficient estimation below:
>
> The use of C-VAE to approximate the distribution of offline datasets is widely adopted in offline RL [1]. Its ability to handle out-of-distribution (OOD) samples is generally considered reliable. To support this, we evaluated the C-VAE’s capability to distinguish in-distribution (ID) and OOD samples. Specifically, we collected ID and OOD samples from our offline environments and conducted the following experiments:
>
> - (i) We computed the corresponding $p(s,a)$ values for each sample, as shown in Appendix C.6, Fig. 5. The results indicate that the $p(s,a)$ distributions of ID and OOD samples are largely non-overlapping, indicating that $p(s,a)$ can effectively separate ID from OOD data.
>
> - (ii) Following the reviewer’s suggestion, we also visualized the latent vectors using t-SNE. The results show that OOD samples are generally close to ID samples in the embedding space, with only a few points far away, further demonstrating the VAE’s strong modeling and estimation ability.
>
> We have added these discussions in Appendix C.6 to better improve clarity.
>
> >**Weakness 1: Tips on controlling the threshold value**
>
> We would like to clarify that the impact of this threshold hyperparameter on the performance is both systematic and interpretable, which makes it straightforward to tune in practice. Moreover, one can pre-compute the distribution of $p(s,a)$ on the offline dataset and select an appropriate threshold accordingly. This procedure requires no additional training or architectural adjustments, provides an accurate estimate of the operating range, and takes only a few minutes to run. Moreover, our experiments consistently show that setting the threshold around 0.6 yields stable and strong performance across environments.
>
> >**Weakness 2.1: experimental setup clarification**
>
> Thank you for your valuable comments. We would like to clarify the experimental setup in Section 5.2 as follows:
>
> - SAMG-even: “1/2 for offline and 1/2 for online” indicates that in the critic mixing step, the offline and online critics are combined with equal weights (0.5 each) for each state-action pair. This corresponds to an ablation of SAMG with a fixed, uniform mixing coefficient instead of the learned state-action-conditional coefficient.
>
> - SAMG-random: For this variant, the mixing coefficient for each state-action pair is randomly drawn from a uniform distribution $p^{off}(s,a) \sim Uniform(0,1)$
>
> - Environment: The experiments in Sec. 5.2 use the averaged D4RL medium-replay environments. We will update the manuscript to explicitly include these details in the experimental setup.
>
> >**Weakness 3: Fig. 1 can provide comprehensive view of the overall architecture**
>
> Figure 1 is designed to be simple and clean, aiming to illustrate all the key components of SAMG and the relationships between them. Although it may appear abstract, we believe it effectively conveys the core structure in a concise manner. We appreciate the feedback and will enhance the figure by adding more explanatory annotations and enriching the interaction between components to better reflect the workflow of SAMG.
>
> >**Question 1: Notation of $\alpha$ in L257**
>
> Thank you for your detailed comments. Notation $\alpha$ in L. 257 should be $p^{off}(s,a)$, and we have already corrected this notation in the manuscript.
>
> >**Question 2: Highlight best scores in Table 1**
>
> Thank you for your valuable suggestion. We have highlighted the algorithm with the best performance in each environment in Table 1 to enhance the clarity of the article.
>
> >**Question 3: Insufficient caption of Fig. 3**
>
> We agree that explicitly clarifying the overlapping curves in the caption would improve clarity, and we have updated the manuscript accordingly.
>
> >**Question 4.1: Curves of $p^{off}(s,a)$ during online fine-tuning**
>
> Thank you for the insightful comment. Our analysis shows that during online fine-tuning, the state–action-conditional coefficient for the offline critic, $p^{off}(s,a)$, gradually decreases over time across different dataset qualities. This indicates that the agent increasingly encounters OOD samples and progressively relies more on the online critic. We have added the corresponding figure to the revised manuscript (Fig. 7-8) to clearly illustrate this trend.
>
> >**References**
>
> [1] Levine et al. Offline reinforcement learning: Tutorial, review, and perspectives on open problems. arXiv 2020.

---

> > ### Comment · Reviewer_vCJH · 2025-11-26
> >
> > Thank you for the supplementary experiments and justifications. In Appendix C.6, Figure 6 (a corresponding caption is wrong. Please correct the caption in the revised manuscript.) helps clarify that C-VAE produces predictable and reliable outputs on both ID and OOD samples. In addition, Figure 5 (including Table 4) explains that the choice of $p(s,a)$ is task-independent and reasonable. These extra results effectively address my concern (W1) on the reliability of C-VAE. Furthermore, I appreciate the authors' efforts in resolving Q1~4 (In Appendix C.8, the description about curve experiments is different with Figure 7; L1178 denotes *pen-cloned* while the figure describes *door-cloned*). Overall, the authors have successfully addressed most of my concerns in the response.
> >
> > However, the authors' response on W2-2, conflicting arguments in L205 and L419 regarding the extremely small value of $p_m^\text{off}$, is omitted in the above response. I will raise the score if the authors provide proper explanations on these or correct my potential misunderstanding.
> >
> > Last but not least, I wonder how the performance changes if the threshold value $p_m^\text{off}$ is adapted alongside the adaptive tuning of C-VAE. By skimming the other reviewers' comments, I have found that replacing the hard cutoff with soft scheduling of $p(s,a)$ incurs a minor performance change. According to the authors' response and Appendix C.6, C-VAE and online critic will progressively capture more accurate and reliable predictions over the online fine-tuning stage. Inspired by Figure 5, I conjecture that the corresponding value of $p(s,a)$ calculated from the updated C-VAE would change simultaneously. In this situation, what will happen if the value of $p_m^\text{off}$ is adaptively updated by computing the empirical upper boundary value $\hat{p_m^\text{off}}$ with the "mastered OOD samples". For instance, one could set the new threshold value $p_m^\text{off}=\hat{p_m^\text{off}}$ if $\max(\hat{p_m^\text{off}})>p_m^\text{off}$ to reflect the updated knowledge into not only the C-VAE and online critic, but also the threshold value. If the remaining time for the discussion is enough to execute further experiments, I would be grateful the empirical results on this experiment. If it is unavailable, I wonder the authors' opinion on using adaptive fine-tuning strategy for the threshold value.

---

> > > ### Author Response · Authors · 2025-11-26
> > >
> > > We sincerely thank the reviewer for the careful reading and for pointing out the issues that we unfortunately missed in the previous response. We also appreciate the helpful comments on the caption mismatch (Appendix C.6 and Figure 7), which have now been corrected in the revised manuscript. Below we address the remaining concern (W2.2) and the new question regarding the adaptive threshold.
> > >
> > > >**W2.2: Conflict between L205 and L419 regarding extremely small $p^{off}_m$**
> > >
> > > We sincerely apologize for overlooking this comment in our earlier response.
> > > The reviewer is absolutely correct that the statement around L419 in the original manuscript was inaccurate. When $p^{off}_m$ is set to small values, all samples satisfy $p^{int} > p^{off}_m$. Therefore, all samples are classified as in-distribution samples, instead of OOD as stated in L 419. p_m^offhas no effect in this extreme case, and the resulting curves will naturally overlap, as observed in Figure 5.
> > >
> > > This is not consistent with the misleading sentence in L419, and we have corrected the logic in the revised manuscript. We thank the reviewer again for catching this issue.
> > >
> > > > **Adaptive tuning of the threshold $p^{off}_m$**
> > >
> > > We appreciate the reviewer’s insightful idea of adapting $p^{off}_m$ jointly with the progressive tuning of C-VAE and the online critic. We fully agree that dynamically adjusting the threshold can in principle provide a more accurate boundary as training evolves.
> > >
> > > However, we slightly disagree with the idea of updating the threshold solely based on the mastered OOD samples, because doing so may push the threshold to an undesirably high value and cause some ID samples from the offline dataset to be mistakenly treated as OOD.
> > >
> > > In fact, we previously experimented with this idea and our procedure was: After each update of the C-VAE, we collected C-VAE outputs on the offline dataset plus the set of mastered OOD samples, fitted the updated empirical distribution, and recomputed a new $p^{off}_m$ based on the distribution.
> > >
> > > Empirically, we observed that the recomputed threshold changed only minimally throughout fine-tuning. This stability indicates that the initially estimated threshold is already a good approximation and that a fixed threshold works reliably in practice. Therefore, we opted for using a fixed value, which simplifies the algorithm, avoids additional computation overhead, and still provides accurate and stable performance.
> > >
> > > We are reproducing this experiment and will update the results as soon as possible. We have added a discussion of this in the revised manuscript in Appendix C.5 to improve clarity.

---

> > > > ### Comment · Reviewer_vCJH · 2025-11-27
> > > >
> > > > Thank you for the subsequent response on the remaining concerns. I appreciate the additional discussions on the adaptive tuning strategy of $p_m^\text{off}$. I will wait for the experimental results and increase the score accordingly.

---

> > > > > ### Author Response · Authors · 2025-11-28
> > > > >
> > > > > We sincerely thank the reviewer for the patience and for taking the time to carefully read and acknowledge our previous response. We greatly appreciate your constructive comments, which helped us refine our analysis.
> > > > >
> > > > > We have completed the experiments regarding the adaptive tuning of the threshold $p_m^{off}$, as previously mentioned, and summarize the findings as follows:
> > > > >
> > > > > We ran experiments on three representative environments: Walker2d-medium, Antmaze-medium-play, and Pen-cloned. Each result is averaged over three random seeds. In these experiments, we dynamically updated $p_m^{off}$ (referred to as SAMG-update), and its values changed over training as shown below (see the full figure in Appendix C.5).
> > > > >
> > > > > |Train steps| 0 | 50k | 100k |150k | 200k |
> > > > > |-|-|-|-|-|-|
> > > > > |Walker2d-medium|0.600|0.603|0.607|0.602|0.605
> > > > > |Antmaze-medium-play|0.600|0.596|0.601|0.609|0.608
> > > > > |Pen-cloned|0.600|0.605|0.611|0.607|0.606
> > > > >
> > > > > The performance comparison between updating and not updating $p_m^{off}$ is as follows:
> > > > >
> > > > > |Algorithm| SAMG | SAMG-update
> > > > > |-|-|-|
> > > > > |Walker2d-medium|88.6|88.7
> > > > > |Antmaze-medium-play|81.4|81.1
> > > > > |Pen-cloned|106.0|106.2
> > > > >
> > > > > From these results, we observe that the changes in $p_m^{off}$ due to dynamic updating are negligible, and the performance gains from updating are minimal. This confirms that using a static $p_m^{off}$ is reasonable, accurate, and compute-efficient.

---

### Official Review · Reviewer_NVzH · 2025-11-03

**Soundness:** 2
**Presentation:** 3
**Contribution:** 2
**Rating:** 4
**Confidence:** 4

**Summary:**

The paper proposes SAMG, an offline-to-online RL fine-tuning framework that fuses a frozen offline critic with the online Bellman target using a state–action probability gate estimated by a conditional VAE. The gate emphasizes the offline critic in in-distribution regions and suppresses it for out-of-distribution samples, is updated during training, and comes with convergence-style arguments.

**Strengths:**

No longer relies on offline data replay; online sample efficiency is high. The formulation is simple and easy to plug into various Q-learning fine-tuning methods.

With C-VAE plus adaptive updates, p can gradually expand the “mastered OOD regions” as training proceeds, and the paper provides clear implementation details.

**Weaknesses:**

1. The design makes several simplifications that may harm performance by introducing imprecision. First, it models the encoder outputs as independent Gaussians and linearly weights them, ignoring correlations and richer distributional structure. Forcing (p) below a threshold directly to zero (Eq. (5)) is too hard, causing non-differentiable/discontinuous use of information and sensitivity to training noise. The CQL extra penalty is altered by adding an “offline version” weighted by (p); the authors admit this “slightly deviates” from the original setup, implying less-fair comparisons and potential divergence risk.
2. The theoretical assumptions are strong. The accelerated-convergence bound depends on the ratio of offline/online sub-optimality bounds, which is unobservable in practice, and the assumption that the offline bound is significantly tighter may not always hold.
3. The O2O fine-tuning based on CQL/AWAC/IQL is not specifically optimized for the online phase. As noted in related work, “a series of Q-ensemble-based algorithms are proposed, combined with balanced experience replay.” The authors should discuss more thoroughly to demonstrate broader effectiveness and performance advantages of the approach.
4. Notational issues: please carefully check the entire paper. In Equation (2), the KL divergence is written as Enc (|) Dec, which seems unreasonable—the prior should be (p(z)), not the decoder. Also, the state–action-conditional coefficient (\alpha) should be consistent with (p(s,a)); if I’m not mistaken, this should be clarified.

**Questions:**

see weaknesses above.

---

> ### Author Response · Authors · 2025-11-25
>
> The reviewer notes the strong online sample efficiency, the simple and easily pluggable formulation, and the clear C-VAE–based adaptive expansion into OOD regions. We respond to all comments point by point below.
>
> **Weakness 1: Simplifications may harm performance**
>
> Thank you for your insightful comments. We believe these mentioned choices aim to balance efficiency, stability, and performance in the O2O setting.
>
> - We model encoder outputs as conditionally independent Gaussians and linearly weight them for computational efficiency and stability. Although this ignores some correlations, our C-VAE estimates the state-action-conditional coefficient $p(s,a)$, capturing the distribution’s complexity (Sec. 3.1 and App. C.2). Experiments on D4RL benchmarks show this simplification does not harm performance.
>
> - We experimented with soft schedules instead of hard cutoff, denoted as SAMG-S, which performed slightly worse in our standard benchmarks, as shown below. The hard cutoff simplifies implementation and ensures that OOD samples with very low probability do not dominate updates. Details are added in Appendix C.7 to improve the clarity of our work.
>
> |Environments| Hopper-medium-replay | Hopper-medium | Antmaze-medium-diverse| Antmaze-medium-play |
> |:-:|:-:|:-:|:-:|:-:|
> | IQL |86.2|62.1|76.4|76.2
> | SAMG|100.4|68.4| 96.6|95.2
> | SAMG-S|98.0|67.4|94.8|92.0
>
> - Modified CQL penalty: In our O2O framework, we adjust a single hyperparameter in the CQL penalty within our improved algorithm. Importantly, the underlying CQL still uses the paper-recommended optimal hyperparameters. This adjustment is intended to balance offline stability and online flexibility.
>
> Overall, these design choices are a deliberate trade-off. Our theoretical analysis and experiments demonstrate that SAMG effectively leverages offline and online data in O2O RL.
>
> **Weakness 2: Theoretical assumptions**
>
> Thank you for your valuable comments. We respectfully clarify the role and practical relevance of the assumptions below:
>
> - The assumption that the offline sub-optimality bound is significantly tighter corresponds precisely to the in-distribution (ID) case, where the offline dataset already provides a well-estimated critic over a substantial portion of the state–action space. This condition is not intended to hold universally, but it is realistic and practically meaningful.
>
> - Our theoretical result is meant to explain why leveraging offline value information can accelerate online learning. The ratio of offline/online sub-optimality bounds appears in the theory purely as an analytical device to characterize this effect; the algorithm does not require estimating this quantity in practice.
>
> - In the empirical evaluation, we observe that SAMG consistently improves sample efficiency. This indicates that the method is robust beyond the idealized assumption, and the theory should be interpreted primarily as a qualitative explanation rather than a restrictive requirement.
>
> We will revise the manuscript to clarify the sufficiency of the assumption, and to add a discussion on its practical interpretation.
>
>
> **Weakness3: Baselines are not specifically designed for O2O**
>
> Thank you for the valuable comment. We chose CQL/AWAC/IQL as O2O baselines because they are widely used and representative offline-to-online strategies, and they provide the most direct point of comparison to our method. Many prior works have also used these methods as the basis for comparisons [1-2], which further justifies our choice.
>
> Moreover, we have compared SAMG with a variety of other state-of-the-art baselines, including Cal-QL, EDIS, and WSRL. These experiments collectively demonstrate the effectiveness and broad applicability of SAMG.
>
> **Weakness3: Discussion of ensemble-based O2O baselines**
>
> We would like to clarify that we recognize the importance of ensemble-based methods and have already included ensemble-based comparisons in our evaluation. Specifically:
>
> - Table 1 reports results against the ensemble regularization method EDIS.
>
> - Appendix D.7 provides an extensive comparison with the ensemble-based baseline WSRL, showing that our method consistently matches or outperforms it.
>
> - Appendix D.7 further shows that combining our approach with ensemble-based techniques yields additional improvements, indicating that our method is complementary to ensemble strategies.
>
> **Weakness 4: Notational issues**
>
> Thank you for your detailed suggestion. We have reviewed the entire manuscript and made the following revisions:
>
> - We confirm that the KL divergence should be written as $KL(q_{Enc}(z|s, a) || p_{prior}(z|s,a))$ rather than in the previously written form involving the decoder.
>
> - We unify the notation for state-action-conditional efficient to $p^{off}(s,a)$ and modify the notation $\alpha$ in Line 258.
>
> We have also checked other parts of the manuscript and corrected some notations that could potentially cause misunderstandings.

---

> > ### Comment · Reviewer_NVzH · 2025-11-27
> >
> > Thank you for the detailed rebuttal and the corresponding revisions. I appreciate the effort to clarify both the methodological design choices and the empirical behavior of SAMG.
> >
> > On the positive side, the additional experiments and discussions do address several of my earlier concerns:
> >
> > - The broader comparison makes the positioning of the approach in the O2O RL landscape clearer and shows that the proposed scheme is reasonably robust across diverse base algorithms.
> >
> > - The empirical study of the hard vs. soft thresholding of the state–action–conditional coefficient, and the ablation with SAMG-S, help justify the specific choice of a hard cutoff. While this remains a heuristic design, the reported results indicate that it is stable and slightly more effective in the tested settings.
> >
> > - The clarification of the theoretical assumptions, and the explicit statement that the tabular convergence-style analysis is meant as a qualitative explanation rather than a practical guarantee in deep function-approximation regimes, are helpful for interpreting the theory.
> >
> > - The notational issues I pointed out (e.g., the KL term in Eq. (2) and the consistency of the state–action–conditional coefficient notation) have been fixed, which improves readability.
> >
> > At the same time, some of my original reservations remain only partially resolved. In particular:
> >
> > Several key design choices — diagonal Gaussian encoder with linear weighting, the hard thresholding of the coefficient, and the modified CQL penalty with an additional “offline” term — are still largely justified empirically rather than from first principles. The new experiments make me more confident that these heuristics behave well on the reported benchmarks, but they also highlight that SAMG trades some theoretical neatness for practicality, and it would be valuable to better isolate the impact of these choices (e.g., in terms of stability and fairness to the base algorithms).
> >
> > The theoretical analysis continues to rely on relatively strong assumptions (such as the offline sub-optimality bound being significantly tighter than the online one) and on a tabular setting. This is acceptable as a stylized analysis, but it limits how far the guarantees can be taken as evidence for behavior in realistic deep RL scenarios.
> >
> > Overall, the rebuttal and added results improve my understanding of the method and increase my confidence that SAMG is a practical and generally beneficial offline-to-online wrapper. However, I still view the conceptual novelty as moderate and the design as somewhat heuristic, with theory that is informative but not fully aligned with the deep RL setting used in experiments. I therefore keep my overall score and recommendation.

---

> > > ### Author Response · Authors · 2025-12-03
> > >
> > > We sincerely thank the reviewer for the thoughtful second-round comments.
> > > We are encouraged that the reviewer confirms that our rebuttal substantially improved clarity, strengthened justification, and addressed the major technical concerns raised in the initial review. As for further concerns, we respond to them point by point as follows:
> > >
> > > >**Principles of key design choices**
> > >
> > > We would like to kindly clarify that the diagonal Gaussian encoder with a linear head is fully consistent with the **principles of VAE**, whereas the other two components—the modified CQL penalty and the hard threshold on the coefficient—do **not** involve any first-principle considerations.
> > >
> > > Specifically,
> > >
> > > - diagonal Gaussian encoder with linear weighting: The diagonal Gaussian encoder directly aligns with the **VAE mechanism** of mapping complex data distributions to a simple Gaussian distribution. We use this capability to quantify "offline degree" of a sample. Regarding the linear weighting, given that the VAE encoder is already expressive and nonlinear, a simple linear weighting is sufficient to combine the online and offline critics, as shown in Sec. 4.2.
> > >
> > > - Hard threshold. The soft and hard thresholding strategies are both **experimental choices**. Our comparison shows that the hard threshold is simpler and slightly more robust in practice.
> > >
> > > - Modified CQL penalty. We would like to re-clarify that **No new hyperparameters are introduced**. We only reduce the existing hyperparameter $\alpha$ in the original CQL algorithm to ensure appropriate offline guidance. Thinking $\alpha$ naturally **varies** across environments in standard CQL practice, adjusting this empirically tuned parameter is fully reasonable and is not tied to any first-principle justification.
> > >
> > > >**Theoretical analysis**
> > >
> > > We would like to clarify that the statement “the offline sub-optimality bound is significantly tighter than the online one” is **conditional**, rather than universally holding as suggested. In fact, Eqn. (9) explicitly clarifies that this bound holds only for states and actions that are well captured by the offline model. Therefore, this assumption is consistent with its definition and reasonable within the stated conditions.

---

### Official Review · Reviewer_XYPj · 2025-11-03

**Soundness:** 3
**Presentation:** 3
**Contribution:** 3
**Rating:** 4
**Confidence:** 4

**Summary:**

The paper proposes SAMG, a simple way to do offline-to-online RL without keeping the offline dataset during fine-tuning. It keeps the pre-trained offline critic and blends its guidance into the online value updates using a state–action–aware weight that estimates how in-distribution each sample is (learned with a conditional VAE). As training progresses, this weight (and, if useful, the offline critic) are updated so reliance on offline guidance fades where the agent has learned. Plugged into CQL, AWAC, and IQL, SAMG improves returns and especially early online sample efficiency on standard benchmarks. The theory supports stability for policy evaluation.

**Strengths:**

- Removes the need to keep the offline dataset during fine-tuning by guiding updates with a frozen offline critic $Q_{\text{off}}$, so it's practical and easy to plug into standard Q-learning methods.
- Uses a state–action–conditional weight $p(s,a)$ so guidance is per-sample (high when in-distribution, low when OOD)
- The weighting model is adaptive during fine-tuning: as the policy improves, reliance on $Q_{\text{off}}$ fades.
- Works as a drop-in wrapper for multiple baselines (e.g., CQL, IQL, AWAC) and consistently improves early online sample efficiency and final returns on standard O2O benchmarks.
- Clear intuition via the intrinsic-reward view

**Weaknesses:**

- Theoretical guarantees cover policy evaluation; there is no convergence guarantee for control with function approximation.
- The $p(s,a)$ pipeline is complex/heuristic which may affect robustness across domains.
- The adaptive update relies on low online Bellman error to tag “mastered” OOD samples and can periodically replace the “offline” critic with the current $Q$; this feedback loop may drift or mis-label samples.
- Compute reporting is light: claims of minimal overhead aren’t backed by detailed wall-clock or memory breakdowns during online fine-tuning.
- Performance is uneven in some tasks and depends on the base algorithm and the quality of the offline critic
- Slight ambiguity around “offline-data-free” when re-creating certain offline regularisers

**Questions:**

1. Why the specific construction of $p(s,a)$ from the C-VAE encoder instead of simpler OOD scores (e.g., reconstruction error) or density-ratio/score-based estimates?
2. What happens if you remove the hard cutoff and use a soft schedule for $p(s,a)$ below $p_m^{\text{off}}$, does this help on OOD-heavy tasks?
3. How sensitive is performance to the offline pretraining budget and to the adaptive update cadence/thresholds?
4. Please provide wall-clock and peak memory comparisons during online fine-tuning, with an explicit breakdown of the C-VAE update cost and frequency.
5. Clarify how the “offline” regularisation terms that reference $\mathcal D$ are implemented without access to $\mathcal D$ during fine-tuning. If a learned behaviour model substitutes for $\mu(a\mid s)$, how is its error controlled?
6. Can you report a failure analysis in settings where SAMG underperforms and Bellman error over time

---

> ### Author Response · Authors · 2025-11-25
>
> The reviewer emphasizes the practicality of removing offline data replay, the per-sample guidance via
> $p(s,a)$, the adaptive fading of offline reliance, the drop-in improvements across multiple baselines, and the clear intrinsic-reward intuition. We address these concerns with two parts.
>
> # Part 1
>
> >**Weakness 1: Setting of the theoretical analysis**
>
> Thank you for your insightful comments. We would like to clarify that our analysis in the tabular setting provides a clear and theoretically grounded foundation for understanding the core mechanisms of SAMG. Moreover, many prior works have conducted convergence analyses under similar tabular assumptions [1].
>
> We will include a discussion in the Limitations section to explicitly state that our theoretical guarantees are based on the tabular setting and currently do not extend to more complex function approximation frameworks. We consider this an important direction for future work.
>
> >**Weakness 2: The $p(s,a)$ pipeline is heuristic**
>
> Thank you for your insightful comments. We respectfully clarify that multiple components in the $p(s,a)$ pipeline are designed for practical and reliability:
>
> - The state–action-conditional coefficient is learned via C-VAE, which is widely used in offline RL to approximate the distribution of offline datasets [2], and its training process is well-established.
>
> - In practice, this design is robust across different benchmarks, and SAMG illustrates consistent performance improvement compared to vanilla baselines, as shown in Sec. 5.
>
> >**Weakness 3: Potential drift or mis-labeling in adaptive updates**
>
> Thank you for your valuable question. We would like to clarify that this concern does not arise in practice.
>
> - The C-VAE structure effectively separates in-distribution (ID) from out-of-distribution (OOD) samples, so high-uncertainty or OOD samples naturally receive low weights. To prove this, we collected ID and OOD datasets from some offline environments and evaluated the corresponding $p(s,a)$ values, as shown in Appendix C.6. Fig. 5. The results show that the $p(s,a)$ distributions of ID and OOD samples are largely non-overlapping, indicating that C-VAE effectively distinguishes ID from OOD.
>
> - In practice, we consistently observe that this mechanism yields stable and robust performance across tasks, without introducing adverse effects, as shown in Fig. 2. This empirical behavior further supports the effectiveness of the proposed paradigm.
>
> >**Weakness 4 & Question 4: Compute reporting and memory breakdowns**
>
> Thank you for the detailed question. We have added new experiments reporting wall-clock time and memory usage across different algorithms. To isolate the computational cost of the C-VAE module, we additionally introduce a variant SAMG-V, which removes the C-VAE component while keeping all other parts identical. All results are averaged across 200k iterations.
>
> |Model| IQL | SAMG-V | SAMG| EDIS|
> |:-:|:-:|:-:|:-:|:-:|
> | Training Speed (per iteration) |0.12s|0.18s|0.19s|1.06s
> | Peak GPU Memory during Backward|286MB| 337MB|361MB |1672MB
>
> The results show that SAMG introduces only minor memory overhead and negligible slowdowns in training speed. In contrast, other approaches such as EDIS introduce substantial memory usage and significantly slower online fine-tuning. These comparisons collectively demonstrate that SAMG remains lightweight in practice, and that the VAE module adds only marginal computational cost.
>
> >**Weakness 5: Performance depends on the base algorithms**
>
> SAMG acts as a wrapper around a base O2O RL method, so its performance naturally depends on the underlying algorithm, which is consistent with prior O2O literature [3-4].
>
> Importantly, because these base algorithms exhibit highly diverse behaviors, the fact that SAMG provides consistent improvements across all of them, further demonstrates the robustness and general effectiveness of the SAMG paradigm.
>
> >**Weakness 6: Ambiguity around “offline-data-free**
>
> We respectfully clarify that SAMG does not access the offline dataset during fine-tuning and the C-VAE module is pre-trained offline as well. All computations during online fine-tuning rely solely on the critic models and the online buffer, never on the original offline data.

---

> ### Author Response · Authors · 2025-11-25
>
> # Part 2
>
> >**Question 1: Why $p(s,a)$ is constructed from C-VAE**
>
> We chose the C-VAE encoder because it provides a probabilistic, state–action-aware embedding, which is more informative than simple reconstruction error or density-ratio estimates. It is widely used in offline RL to approximate the distribution of offline datasets [2], and its training process is well-established.
>
> >**Question 2: Hard cutoff v.s. soft schedule for $p(s,a)$**
>
> We experimented with soft schedules instead of hard cutoff, denoted as SAMG-S, which performed slightly worse in our standard benchmarks, as shown below. The hard cutoff simplifies implementation and ensures that OOD samples with very low probability do not dominate updates.
>
> |Environments| Hopper-medium-replay | Hopper-medium | Antmaze-medium-diverse| Antmaze-medium-play |
> |:-:|:-:|:-:|:-:|:-:|
> | IQL |86.2|62.1|76.4|76.2
> | SAMG|100.4|68.4| 96.6|95.2
> | SAMG-S|98.0|67.4|94.8|92.0
>
> Details are added in Appendix C.7 to improve the clarity of our work.
>
> >**Question 3: Sensitivity to offline pretraining budget**
>
> Thank you for your valuable feedback. We varied the size of the offline pretraining dataset to 100% (full), 75%, and 50% of the standard dataset. In the table below, variants with reduced pretraining data are denoted as CQL-75/SAMG-75 and CQL-50/SAMG-50, corresponding to 75% and 50% of the original offline dataset, respectively. The results show that while reducing offline data slightly decreases performance for both CQL and SAMG, our SAMG consistently outperforms CQL across all settings.
>
> |Method|CQL|SAMG|CQL-75|SAMG-75|CQL-50|SAMG-50|
> |:---:|:---:|:---:|:---:|:---:|:---:|:---:|
> |Halfcheetah-medium|47.6|59.0|42.1|55.6|38.0|52.7|
> |Halfcheetah-medium-expert|95.2|97.2|90.5|95.3|84.1|93.2|
> |Pen-cloned|90.0|96.2|68.1|86.0|43.9|77.4|
> |Door-cloned|-0.34|70.8|-0.42|61.5|-3.18|54.9|
>
>
> >**Question 5: Sensitivity to Update Cadence**
>
> We also investigated the impact of the adaptive update frequency, varying the cadence of online updates to every 1k, 5k, 10k, and 20k steps. In the table below, SAMG-1k, SAMG-5k, SAMG-10k, and SAMG-20k indicate the corresponding update frequencies. Results indicate that SAMG’s performance is relatively stable across different update frequencies, suggesting robustness to the choice of adaptive update frequency.
>
> |Method|AWAC|SAMG-1k| SAMG-5k |SAMG-10k|SAMG-20k|
> |:---:|:---:|:---:|:---:|:---:|:---:|
> |Walker2d-medium|87.8|103.4|105.8|103.6|99.5|
> |Antmaze-umaze|103.6|88.2|86.2|87.0|84.6|
>
> >**Question 6: Failure analysis where SAMG underperforms**
>
> Thank you for the suggestion. In our experiments, we did not observe clear failure cases where SAMG underperforms the base algorithm. Across all tasks and across different offline critics, SAMG consistently provides improvements or remains comparable with the base method.
>
> To further verify this, we monitored the Bellman error of both the offline and online critics over the entire fine-tuning process. The Bellman error does not exhibit instability spikes or divergence patterns under SAMG, which validates the stability of SAMG.
>
> > **References**
>
> [1] Wen, et al. Characterizing the Gap Between Actor-Critic and Policy Gradient. ICML 2021.
>
> [2] Levine et al. Offline reinforcement learning: Tutorial, review, and perspectives on open problems. arXiv 2020.
>
> [3] Wang et al. Train Once, Get a Family: State-Adaptive Balances for Offline-to-Online Reinforcement Learning. NeurIPS 2023.
>
> [4] Liu et al. Energy-Guided Diffusion Sampling for Offline-to-Online Reinforcement Learning. ICML 2024.

---

### Official Review · Reviewer_1Wb2 · 2025-11-04

**Soundness:** 3
**Presentation:** 2
**Contribution:** 3
**Rating:** 4
**Confidence:** 4

**Summary:**

The paper proposes State-Action-Conditional Offline Model Guidance (SMAG) for offline-to-online reinforcement learning. The approach eliminates the need to retain offline datasets during fine-tuning by leveraging a frozen offline critic to guide online learning. A state-action-conditional coefficient, instantiated using a CVAE, adaptively determines how much to rely on offline or online critics. Theoretical analyses demonstrate convergence speed, and experiments on the D4RL benchmark show superior empirical performance.

**Strengths:**

- The idea of freezing the offline critic and using an adaptive coefficient to guide online updates is well-motivated.
- SAMG outperforms base algorithms (CQL, IQL, AWAC) and advanced baselines such as Cal-QL and EDIS on diverse D4RL benchmark tasks (Mujuco locomotion, AntMaze, Adroit).
- The method is compatible with various Q-function-based algorithms, allowing easy adaptation to existing RL frameworks.

**Weaknesses:**

I am willing to increase the score if my concerns are addressed.

- The presentation could be improved through clearer structure, tighter writing, and better figures and tables. For example,
    - The term $p_{int}$ should be clearly defined before its first use at Line 203, i.e., $p_{int}$ = Eq. (4).
    - The name "state-action-conditional coefficient" could benefit from a simpler, more intuitive description. Similarly, the method name State-Action-Conditional Offline Model Guidance may benefit from a more precise description, as I find the current name somewhat ambiguous.
    - In Figure 1, the upper part should explicitly indicate that the frozen and inherited components correspond to the Q network.
    - Captions for Figures 2-4 are somewhat difficult to read. It might be beneficial to integrate them into a single figure with three subplots, labeled (a) (b) (c).
    - Experimental settings (e.g., hyperparameters, update frequency) could be summarized in a table for clarity.
- The paper lacks some learning curve comparisons, which could better illustrate the differences in performance and learning speed in O2O scenarios than tables alone.
- The experimental analysis would be more informative if the coefficient $p(s,a)$ were visualized to provide insights into the learning dynamics.
- In offline and O2O RL, [1] mixes in-sample and out-of-sample maximum Q-values in the Bellman target to control generalization. Since the idea of mixing two different Q-values in the target is conceptually similar, it would be beneficial to include a discussion in the paper.

[1] Doubly mild generalization for offline reinforcement learning, NeurIPS 2024.

**Questions:**

N/A

---

> ### Author Response · Authors · 2025-11-25
>
> The reviewer highlights the well-motivated use of a frozen offline critic with adaptive guidance, SAMG’s strong performance over both base and advanced baselines, and its easy adaptability to various Q-function–based algorithms. We appreciate the feedback and respond to each comment in detail below.
>
> >**Weakness 1: Presentation improvement**
>
> Thank you for the constructive feedback on presentation. We have carefully revised the manuscript to improve clarity, structure, and figures. Specifically:
>
> - **Definitions and terms**: We now define the term $p_{int}$ clearly in Eq. 4. We will simplify the description of “state-action-conditional coefficient” to “offline weight” for better interpretability.
>
> - **Figures**: In Figure 1, we now explicitly indicate that the frozen and inherited components correspond to the Q network. Captions for Figures 2–4 have been improved for readability, and we have combined them into a single figure with three labeled subplots.
>
> - **Experimental settings**: We have summarized all key hyperparameters, including update frequencies, and other experimental details in Table 4. to improve clarity and reproducibility
>
> >**Weakness 2: Training curve comparisons**
>
> We have added learning curve comparisons in Appendix D.8 to better illustrate the differences in performance, which illustrates that SAMG **converges much faster** (requiring only about 20% of the training steps used before) and achieves **better final performance**. We will organize more results and update them.
>
> >**Weakness 3: Visualization of $p(s,a)$**
>
> Thank you for your insightful comments. We conducted experiments to include visualization of $p(s,a)$, including both quantitative comparisons between in-distribution (ID) and out-of-distribution (OOD) data, as well as curves of $p(s, a)$ throughout the training.
>
> Specifically, we collected ID and OOD datasets from our offline environments and evaluated the corresponding $p(s,a)$ values, as shown in Appendix C.6, Fig.5. The results show that the $p(s,a)$ distributions are largely non-overlapping, suggesting that $p(s,a)$ effectively distinguishes ID from OOD.
>
> We further provide the curves of $p(s,a)$ through the training process. As shown in Fig. 7-8, the coefficient naturally transitions from higher to lower values over training, reflecting a shift from ID regions to OOD regions. This visualization confirms the intended learning dynamics and further supports the design of SAMG.
>
> Details are provided in Appendix C.6 in the manuscript to improve the clarity of our work.
>
> >**Weakness 4: Discussion with offline RL algorithm DMG**
>
> Thank you for the helpful suggestion. We carefully reviewed DMG [1] and agree that it is conceptually related. However, our approach differs from DMG in both objective and mechanism:
>
> - Where the mixing happens: DMG mixes value estimates within the Bellman target, whereas SAMG mixes a frozen offline critic with the online critic at the model level, using a learned state–action conditional coefficient to adaptively control the contribution of offline knowledge.
>
> - Intended setting: DMG focuses on value generalization in offline RL, while SAMG is designed for O2O adaptation, aiming to improve online sample efficiency without retaining the offline dataset during fine-tuning.
>
> - Theoretical focus: The assumptions and analytical goals differ, so the theoretical results are not directly comparable.
>
> We have incorporated a discussion of DMG in the revised Related Work section to clarify both the conceptual connection and the methodological distinctions.
>
> >**References**
>
> [1] Mao et al. Doubly mild generalization for offline reinforcement learning, NeurIPS 2024.

---

### Official Review · Reviewer_xnw6 · 2025-11-04

**Soundness:** 3
**Presentation:** 3
**Contribution:** 2
**Rating:** 4
**Confidence:** 3

**Summary:**

This paper proposes SAMG, which uses a pre-trained offline critic model for Q-value updates in online reinforcement learning, removing the need for an offline dataset. Essentially, SAMG introduces an intrinsic reward term proportional to the difference between the offline and online Q-values. Experiments demonstrate the effectiveness of the proposed method.

**Strengths:**

- The proposed method is clearly present and easy to follow.
- The experiments are fairly extensive.

**Weaknesses:**

- Using a Conditional VAE to estimate $p(s, a)$ plays a crucial role in the algorithm's performance, but the paper lacks sufficient evidence that the Conditional VAE can accurately measure the degree of OOD in a data point. The authors should provide more experimental results to support this claim.
- The intrinsic reward $r^{in}(s,a) = \gamma p(s,a) Q^{off}(s',a') - Q(s',a')$ seems questionable. If the online Q is initialized from the offline Q, then within the offline data distribution $Q^{off}(s',a')$ and $Q(s',a')$ will be close, making $r^{in}(s,a)$ nearly zero. Outside the offline distribution, $p(s,a)$ is small, so $r^{in}(s,a)$ would also approach zero. Overall, it seems that $r^{in}(s,a)$ may have little practical effect. Intuitively, this intrinsic reward may not work as intended.

Overall, I consider this paper borderline. If the authors can clearly address my concerns, I’d be happy to raise my score.

**Questions:**

The intrinsic reward $r^{in}(s,a)$ essentially introduces Q-value information into the reward, allowing the reward to encode some long-horizon signal. This might be the reason why the proposed method achieves better sample efficiency. I'd like to hear the authors’ thoughts on this perspective.

---

> ### Author Response · Authors · 2025-11-25
>
> We sincerely thank the reviewer for recognizing that our method is clearly presented and that the experiments are extensive. We address each concern point by point below.
>
> **Weakness 1: Can C-VAE measure the degree of OOD?**
>
> Thank you for your insightful comments. We conducted experiments to support this claim, including both quantitative comparisons between in-distribution (ID) and out-of-distribution (OOD) data, as well as qualitative analyses of representative samples.
>
> Specifically, we collected ID and OOD datasets from our offline environments and evaluated the corresponding $p(s,a)$ values, as shown in Appendix C.6, Fig. 5. The results show that the $p(s,a)$ distributions are largely non-overlapping, indicating that $p(s,a)$ effectively distinguishes ID from OOD.
>
> We further provide illustrative examples to highlight C-VAE’s ability to distinguish between the two types of data. Details are provided in Appendix C.6 in the manuscript to improve the clarity of our work.
>
> **Weakness 2: Intrinsic reward may be near zero**
>
> Thank you for your valuable question. We respectfully clarify that $r^{in}$ is not always zero in ID conditions and the near-zero intrinsic rewards in OOD conditions are intended behavior. Specifically:
>
> - For the ID condition, although $Q$ is initialized by $Q^{off}$, due to the challenges of O2O training, $Q$ may be significantly affected and thus deviate from the correct value for ID samples. In this case, this term provides a reasonable intrinsic reward and corrects the bias in the Q function.
>
> - For the OOD condition, $p(s, a)$ is close to 0, meaning that it does not provide any potentially misleading guidance under uncertain OOD conditions, which is exactly what we expect.
>
> We have added this part of the discussion to the manuscript to better demonstrate clarity.
>
> **Question 1: Can intrinsic reward explain the sample-efficiency gains and how?**
>
> The intrinsic reward indeed contributes to the sample-efficiency improvements observed in SAMG.
>
> - Accurate guidance: The intrinsic reward in this form can provide correct guidance for ID conditions and avoid misleading information in OOD situations, contributing to more stable and effective learning.
>
> - Long-horizon signal: The intrinsic reward is directly based on the Q-function, offering long-horizon guidance that is also directly grounded in the function itself, offering a more temporally coherent learning signal.
>
> - Moreover, SAMG maintains a replay buffer composed exclusively of online samples, without retaining offline data. This design further enhances sample efficiency.
>
> Together, these components explain the efficiency gains achieved by SAMG.

---

### Author Response · Authors · 2025-12-03
**Global Response**

Dear AC, SAC, PC, and Reviewers:

We sincerely thank all reviewers for their thorough and thoughtful feedback. Each reviewer provided valuable comments and suggestions that helped us improve the manuscript.

We are encouraged that R. 1Wb2 and R. cVJH recognize the idea of **removing the need to retain offline dataset** and **weighting the offline and online target critics with a coefficient well-motivated** (R. 1Wb2) and **simple but effective** (R. vCJH). We also appreciate that R. XYPj and R. vCJH acknowledge the **adaptive updates of the coefficient during online fine-tuning**. Moreover, several reviewers (R. NVzH, R. 1Wb2, R. XYPj, R. vCJH) find the SAMG paradigm **easy to plug into various Q-learning fine-tuning methods**, highlighting its plug-and-play nature and scalability.

We are also pleased that our experiments are viewed as **fairly extensive** (R. xnw6) and **consistently improves early online sample efficiency and final returns** (R. 1Wb2, R. vCJH). R. vCJH further agrees that **the theoretical analysis helps in understanding** SAMG. We additionally appreciate that R. xnw6 finds **the proposed method clearly presented and easy to follow**.

In response to the reviewers’ constructive suggestions, we have resolved all the issues with new experiments, clearer explanations and additional evidence:

- **For visualization of coefficient $p(s, a)$**, we have provided visualization of $p(s, a)$, including distribution (R. xnw6, R. 1Wb2, R. XYPj), t-SNE visualization of ID and OOD data (R. vCJH), and curves of $p(s, a)$ throughout the training. (R. 1Wb2, R. vCJH). These visualizations confirm the reliability of $p(s, a)$.

- **For experiments**, we added experiments on compute reporting and memory breakdowns (R. XYPj) and adaptive tuning of the threshold (R. vCJH). We also clarified the experimental setups in Sec. 5.2 (R. vCJH, R. NVzH), illustrated the training curves of SAMG (R. 1Wb2), and clarified the discusison of ensemble-based O2O baselines (R. NVzH),

- **For intrinsic-reward analysis**, we added more clarification on the values of intrinsic rewards and explained why intrinsic reward can explain the sample-efficiency gains (R. xnw6).

- **For presentation**, we have checked the notational issues in the manuscript, added a comparison in Related Works and updated some figures to enhance the clarity.

SAMG proposes a novel O2O RL paradigm which eliminates the need for retaining offline data and combines offline and online critics with an adaptive coeffcient. SAMG is easy to plug in Q-function-based algorithms and showcases consistent performance improvement. We believe these contributions will have meaningful impact on future offline-to-online RL reasearch.

Finally, we are encouraged that R. vCJH has already engaged in substantial discussion and has expressed a **clear intention to raise the score**. R. xnw6 and R. 1Wb2 similarly noted in their initial comments that **they were willing to increase their scores if the concerns were addressed**. Given that all raised concerns have now been thoroughly resolved with new evidence and clarifications, we sincerely hope that the AC can take these points into consideration when making the final decision.

Best regards,

Authors of submission #17126

---

### Meta-Review · Area_Chair_vgdT · 2026-01-06

**Summary:**

**Paper Summary**

This paper proposes SAMG, a new offline-to-online (O2O) reinforcement learning method that freezes the pre-trained offline critic and uses it as guidance during online fine-tuning, eliminating the need to maintain and retrain on offline datasets. The method employs a state-action-conditional coefficient (learned via C-VAE) to adaptively weight the frozen offline critic with the online target critic. Evaluated on D4RL benchmarks, SAMG achieves full online sample utilization while outperforming existing O2O RL algorithms that require maintaining offline data.

---

After reading the paper, review comments, and author responses, the AC summarizes the paper's strengths and weaknesses below.

**Strengths**
- Motivation: The motivation to remove the dependency on offline data during the online phase is a significant practical advantage for O2O RL methods.
- Method: The "plug-and-play" module design is highly applicable to various existing RL architectures.
- Evaluations: Multiple base methods equipped with the proposed SAMG show consistent performance improvements. The experimental results were conducted over multiple trials to ensure robust evaluation.

**Weaknesses**
- Modeling Capability and Assumptions: Reviewers questioned whether the CVAE can accurately differentiate between in-distribution (ID) and out-of-distribution (OOD) data, especially in high-dimensional spaces where boundaries are often blurred. The use of independent Gaussian distributions for the encoder output was seen as potentially too simplistic to capture the complex, multi-modal nature of real-world state-action data.
- Method Design:
	1. The "hard-switch" design (setting $p(s,a)$ to zero once it falls below a threshold) may cause gradient discontinuity and make the training process sensitive to noise.
    2. The intrinsic reward $r_{in}$ might frequently vanish (become zero) during practical implementation, thereby losing its effectiveness in guiding the agent.
    3. Even with the adaptive guidance design, the method still requires hyperparameters that need manual exploration.
- Evaluations:
	1. While compared with multiple methods, the comparison between the proposed SAMG and SOTA O2O RL methods is lacking. The performance comparison with WSRL in the appendix is only partially presented, and there is no in-depth analysis of convergence speed or model size differences.
    2. Experiments are only conducted in one domain (D4RL). How the proposed method performs in more complex domains remains unknown, such as long-horizon tasks and partially observable environments.
- Paper Writing:The paper should reorganize the content thoroughly and clarify paragraphs with contradictory statements. For example, why does Fig. 7 indicate the latents between ID and OOD are close (where latent embeddings of OOD samples are even closer to the cluster center), while also stating that $p(s, a)$ generated from ID and OOD samples can be separated?

**Reviewer Concerns:**

The concerns raised by the reviewers have been summarized above, including whether the CVAE is oversimplified and whether its ability to distinguish between ID and OOD samples is robust and reliable. After considering the authors' rebuttal, the AC believes that several critical concerns remain unresolved:
1. Limited generalization across domains: Experiments were conducted solely on a single domain. While the AC acknowledges that D4RL is a mainstream benchmark, it remains unclear whether the proposed method can generalize effectively to tasks in other domains.

2. Insufficient justification for method design: The rationale behind certain design choices remains weak. For instance, if referencing state-action pairs is necessary, why were world models or other dynamics models not considered? Furthermore, the assumptions regarding the prior distribution require stronger justification; specifically, why this specific distribution was chosen and whether more suitable alternatives exist.

3. Incomplete comparative analysis with SOTA: The paper lacks a comprehensive comparison and analysis against state-of-the-art (SOTA) methods. Given that removing the dependency on offline datasets is not a novel contribution unique to this work, a more explicit analysis highlighting the fundamental differences between the proposed method and existing approaches is required.

4. Clarity and consistency issues: The overall clarity of the paper needs improvement. In particular, the results presented in Figure 6 and Figure 7 appear to be mutually contradictory and require further clarification.

**Reviewer Scores:**

The paper received initial scores of [4, 4, 4, 4, 4], indicating a borderline consensus leaning toward rejection. Following a detailed rebuttal by the authors, the AC notes that several concerns have been addressed, particularly regarding method design elements like the 'hard-switching' or 'intrinsic reward' mechanism. While the AC acknowledges that some reviewers may be satisfied enough to raise their scores to 6, several critical issues remain unresolved as discussed above. Consequently, the AC recommends rejection but encourages the authors to incorporate the provided feedback to strengthen a future submission.

---

### Decision · Program_Chairs · 2026-01-26

Reject